

# A comparative study of feature selection and feature extraction methods for financial distress identification

Dovilė Kuizinienė[1], Paulius Savickas[1], Rimantė Kunickaitė[1], Rūta Juozaitienė[1], Robertas Damaševičius[1], Rytis Maskeliūnas[2] and Tomas Krilavičius[1]

[1] Department of Applied Informatics, Vytautas Magnus University, Kaunas, Lithuania
[2] Silesian University of Technology, Gliwice, Poland

## ABSTRACT

Financial distress identification remains an essential topic in the scientific literature due to its importance for society and the economy. The advancements in information technology and the escalating volume of stored data have led to the emergence of financial distress that transcends the realm of financial statements and its' indicators (ratios). The feature space could be expanded by incorporating new perspectives on feature data categories such as macroeconomics, sectors, social, board, management, judicial incident, *etc.* However, the increased dimensionality results in sparse data and overfitted models. This study proposes a new approach for efficient financial distress classification assessment by combining dimensionality reduction and machine learning techniques. The proposed framework aims to identify a subset of features leading to the minimization of the loss function describing the financial distress in an enterprise. During the study, 15 dimensionality reduction techniques with different numbers of features and 17 machine-learning models were compared. Overall, 1,432 experiments were performed using Lithuanian enterprise data covering the period from 2015 to 2022. Results revealed that the artificial neural network (ANN) model with 30 ranked features identified using the Random Forest mean decreasing Gini (RF_MDG) feature selection technique provided the highest AUC score. Moreover, this study has introduced a novel approach for feature extraction, which could improve financial distress classification models.

# INTRODUCTION

Financial distress prediction remains an important unsolved issue in the current economy and financial research. Financial distress reflects insufficient cash flow problems arising from the external economic environment or internal financial decisions (*Sun et al., 2020*). An enterprise still has the potential to recover from financial distress. However, becoming a healthy enterprise is more challenging as the period of financial distress increases (*Farooq, Jibran Qamar & Haque, 2018*). This is because it is easier to deal with profitability problems than to fix bankruptcy or liquidity issues. Moreover, the process of bankruptcy or liquidation affects the stability of the business environment (*Zelenkov, 2020*)

Corresponding author
Robertas Damaševičius,
robertas.damasevicius@vdu.lt

and negatively influences stakeholders interested in the performance of the enterprise, *i.e.,* investors, creditors, employees, customers, suppliers, *etc.* (*Cultrera & Brédart, 2016*; *Kumar & Roy, 2016*; *Li & Sun, 2020*). The interest of creditors and investors is obvious—the protection of their own investment (*Inam et al., 2019*). However, financial distress losses can negatively affect the entire society and economy (*Huang et al., 2017*). For instance, a financial crisis correlates with the number of enterprises that experience financial distress at the same time (*Salehi, Mousavi Shiri & Bolandraftar Pasikhani, 2016*). Hence, government institutions are interested in assessing financial distress as one of the economic stability indicators. Reliability is the key aspect of an enterprise from the perspective of the society's interest. Due to the aforementioned reasons, the supplier desires to have assurance that their funds will be refunded, the consumer seeks quality and assurance, and the employee desires to receive their salary on time. The financial distress indicator helps facilitate the decision-making process regarding mutual relations.

Early financial distress indication would help the stakeholders make more accurate decisions, ensuring a healthy enterprise position from a long-run perspective (*Süsi & Lukason, 2019*; *Farooq, Jibran Qamar & Haque, 2018*). In general, it is believed that early warning signals could be detected even before the distress begins (*Xu, Fu & Pan, 2019*; *Huang & Yen, 2019*; *He et al., 2023*). However, the difficulty lies in the high dimensionality of available features. The situation becomes even more complicated due to the visible or hidden relationships among the variables of interest (*Mora García et al., 2008*). The historical background of this matter lies in the financial statement analysis (*Mora García et al., 2008*) relying on derivative indicators, so-called ratios. Most financial ratios are derived from the balance sheet and the income statement, and some require additional market information: P/E, EPS, *etc.* Usually, financial distress studies focus on about 20-30 financial ratios describing companies' state (*Pavlicko, Durica & Mazanec, 2021*; *Liang et al., 2020*; *Shen et al., 2020*; *Cheng, Chan & Yang, 2018*; *Mora García et al., 2008*; *Khoja, Chipulu & Jayasekera, 2019*; *Doğan, Koçak & Atan, 2022*). A more extensive study (*Kim, Mun & Bae, 2018*) included 111 financial ratios. In this case, a two-sample $t$-test was used to identify a subset of 53 statistically significant ratios. Regardless of its rare and delayed occurrence, these studies confirm that the main feature set for financial distress identification comes from the financial statements. Therefore, researchers seek to expand the research field by incorporating additional features, which can be categorized as macro indicators, sector indicators, and additional indicators. Macro indicators include GDP (gross domestic product) growth rate (%), government debt, interest rate, unemployment rate, *etc.* Whereas sector indicators cover NACE code (control indicator), industrial type, GICS Sector - Industrials, industry concentration, industry growth, *etc.* Moreover, additional information consists of enterprise age, type, CEO change, CEO age, CEO criminal record, age diversity, costs of employees, average cost of employees, number of judicial incidences, *etc.* Nevertheless, data enrichment leads to a high-dimensional feature space, which is categorized as big data (*Nath & Kaur, 2019*) due to its vast, varied, and complex structure. Therefore, machine learning and statistical approaches are required to extract interesting and significant patterns for financial distress identification (*Nath & Kaur, 2019*; *Alaminos, del Castillo & Fernández, 2016*). These approaches allow the creation of models with greater

predictive power. However, including many features into a model may lower its accuracy and cause overfitting (*Lohmann, Möllenhoff & Ohliger, 2022*). Thus, the identification of essential features that cause financial distress becomes one of the key factors for powerful model creation (*Ben Jabeur, Stef & Carmona, 2022*).

This study aims to propose a framework for efficient financial distress classification by combining dimensionality reduction and machine learning techniques. To reduce the high-dimensional feature space, 15 different dimensionality reduction methods were tested. Thorough experiments were performed using different subsets of features that were obtained through various feature selection and extraction techniques. The model development phase focused on 17 classification techniques, including supervised and unsupervised machine learning methods. Additional experiments were conducted utilizing a diverse number of hidden layers to identify the most suitable structure for the neural network model. All the experiments presented in this article were performed using the data of 81,202 Lithuanian enterprises covering the period from 2015 to 2022. The analyzed dataset consists of 972 features covering financial records, ratios, macroeconomics and sectors indicators, juridical incidents, *etc*. AUC, Gini and other metrics were used to evaluate the efficiency of the combinations of dimensional reduction and machine learning models.

The article is organized as follows. 'Literature review' presents a literature review on dimensionality reduction techniques in bankruptcy or financial distress prediction. 'Data' describes the Lithuanian enterprise data used in this study. 'Methodology' introduces the proposed theoretical framework. 'Research results' summarizes the obtained results. The final conclusions are discussed in 'Conclusions'.

## LITERATURE REVIEW

The topic of financial distress and bankruptcy has been analyzed since the late sixties, starting with *Beaver (1966)* and *Altman (1968)*. These authors sought to identify key financial ratios capable of predicting financial failure. Even today, financial statements remain the "backbone" for accounting and play an essential role in the decision-making of the stakeholders (*Angenent, Barata & Takes, 2020*). However, researchers are keen on identifying additional features that may indicate financial difficulties, as they are only provided with this information once a year and it takes half a year to obtain it. Other indicators (features) come from the revision of market information, such as macro or sector indicator changes, government's or its related institution's announcements, changes in business management, *etc*.

Technological improvements and data availability has led to incorporation of all these additional features into development of intelligent systems (*Le et al., 2019*). However, increasing the number of features results in negative effects on models. For this reason, data mining and machine learning techniques have recently been employed in finance, mainly to identify the relationship between financial and economic indicators and their predictive capabilities (*Yi et al., 2022*). *Aghakhani et al. (2017)* demonstrates that when addressing the involvement of numerous features with uncertain relationships, the dimensionality

reduction strategy assumes a significant role. Dimensionality reduction methods can be divided into feature selection and feature extraction techniques (see Table 1).

## Feature selection

Feature selection (FS) is a dimensionality reduction approach focusing on removing redundant, noisy and non-informative features from the original feature set (*Al-Tashi et al., 2020*). The implementation of this approach is widespread in the financial distress and bankruptcy concepts due to its possibility to achieve knowledge of the features' importance. Next, examples of feature selection techniques for identifying financial distress are presented. It's worth noting that the majority of studies focused on 3–5 feature selection methods to reveal the best set of features.

Filter methods analyze each feature individually. One of the simplest examples is the correlation feature selection (CFS). The premise of CFS is that a useful feature subset comprises features that exhibit significant correlations with the class (predictive of it), but are not correlated with one another (not predictive of it). The CFS algorithm analyzes and ranks a set of feature subsets, rather than carrying out these steps for a single attribute. The CFS algorithm was analyzed in the Spanish market study (*Faris et al., 2020*), which aimed to improve the accuracy of the classification of bankrupt enterprises. The data was balanced using the SMOTE method, and different FS methods (information gain (IG), Relief-f, correlation Attribute Evaluator (CorrAE), Classifier Attribute Evaluator (CAE)) were applied. Results revealed that CFS achieved the best results, *i.e.,* it achieved G-mean value of 0.66 and AUC of 0.89. The initial dataset contained 33 attributes, of which nine were selected using CFS. Another study (*Séverin & Veganzones, 2021*) attempted to improve the accuracy of bankruptcy prediction by using income information. This study used a similar principle as CFS, *i.e.,* if the attributes had a correlation higher than 0.6, then one of them was removed. Unfortunately, the authors did not analyze the impact of the CFS method on the classification results.

The idea behind the Relief-F method is to assign a weight to each attribute corresponding to its ability to distinguish between different classes. It is assumed that the feature values for the two closest neighbors who belong to the same class have a probability of being the same, while the feature values for the two closest neighbors who belong to different classes have a probability of being different. Researchers (*Faris et al., 2020*) used the Relief-F method, which achieved a 0.63 G-mean value and a 0.91 AUC score. However, further analysis revealed that the Relief-F method was outperformed by CFS and IG methods when used in combination with ML methods. The Relief-F method was also used in another study (*Kou et al., 2021*) that attempted to predict SME bankruptcy cases using transaction data. The authors of this study used the ensemble technique, *i.e.,* the attribute importance index was estimated as a combination of the outputs of Relief-f, Chi-squared, IG and gain ratio (GR) methods. Relief's ability to find features that interact gets worse as more features are added. This happens because the way it calculates neighbors and weights becomes more random as more features are added (*Urbanowicz et al., 2018*).

In statistics, the Chi-square test ($\chi^2$) is used to determine whether two categorical or nominal variables are independent. We can obtain the observed count O and the expected

Kuizinienė et al. (2024), *PeerJ Comput. Sci.*, DOI 10.7717/peerj-cs.1956

**Table 1  Feature selection and extraction literature overview.**

| | Base of comparison | Time complexity | Performance | Suitability on big data | Adavantages | Disadvantages | Methods | References |
|---|---|---|---|---|---|---|---|---|
| Future selection | **Filter** | Low | Low | Very high | Low computational cost, short running time, low risk of overfitting. | Low performance, overlooks feature dependencies, does not interact with the classifier. | CFS, Relief-F, X, K–W, *T*-test. | (*Faris et al., 2020*; *Séverin & Veganzones, 2021*; *Kou et al., 2021*; *Azayite & Achchab, 2018*; *Papíková & Papík, 2022b*; *Ben Jabeur & Serret, 2023*) |
| | **Wrapper** | Medium | High | High | High performance, searches for feature dependencies, interacts with the classifier. | High computational cost, long running time, high risk of overfitting. | Backward wrapper, stepwise LR, stepwise DA, GA, PDC-GA. | (*Perboli & Arabnezhad, 2021*; *Al-Milli, Hudaib & Obeid, 2021*; *Tsai et al., 2021*; *Ben Jabeur, Stef & Carmona, 2022*; *Papíková & Papík, 2022a*) |
| | **Embedded** | Low | Medium | High | Short running time, low risk of overfitting. | Low performance on a small set of features. | LASSO, XGBoost, tree-based, L1-bassed, f-classif, Cat-Boost. | (*Altman et al., 2022*; *Huang, Wang & Kochenberger, 2017*; *Li et al., 2021*; *Du et al., 2020*; *Jiang et al., 2021*; *Papíková & Papík, 2022a*; *Volkov, Benoit & Van den Poel, 2017*; *Zizi et al., 2021*; *Ben Jabeur, Stef & Carmona, 2022*; *Jabeur et al., 2021*; *Doğan, Koçak & Atan, 2022*; *Ben Jabeur & Serret, 2023*) |
| | **Hybrid** | High | High | High | High performance. | Selection dependent on classifier. | Combinations of other methods. | (*Lin & Hsu, 2017*) |
| Future extraction | **Linear** | Low | Medium | Low | Low computational cost, short running time. | Low performance on big data, information loss. | PCA, LDA, LPP, NPE, RSL. | (*Ye, Ji & Sun, 2013*; *Wang, Liu & Pu, 2019*; *Ayesha, Hanif & Talib, 2020*; *Sulistiani, Widodo & Nugraheni, 2022*; *Adisa et al., 2019*; *Jiang et al., 2021*; *Acharjya & Rathi, 2021*) |
| | **Non-linear** | High | High | High | High performance with big data. | High computational cost, long-running time. | MDS, tSNE, SOM, Autoencoder. | (*van der Maaten, Postma & Herik, 2007*; *Mora García et al., 2008*; *Soui et al., 2020*; *Khoja, Chipulu & Jayasekera, 2019*; *Mokrišová & Horváthová, 2020*; *Zoričák et al., 2020*) |

count E from the data of two variables. Chi-square calculates the difference between the expected count E and the observed count O (*Gajawada, 2019*). If the observed count is close to the expected count when two features are independent, then the Chi-square value will be lower. Accordingly, a high Chi-square score suggests that the independence hypothesis is false. In simple terms, features with higher Chi-square values are associated with the response variable and might be selected for model training. The Chi-squared test was used in the previously mentioned study *Kou et al. (2021)* as one of the ensemble techniques. In this case, its influence on the modelling performance was not estimated. The Chi-squared test was also used in combination with classification and regression trees (*Azayite & Achchab, 2018*). However, the authors did not analyze the influence of feature selection on the modelling performance.

Kruskal–Wallis (K–W) and $T$-test are used to determine if there is a statistically significant difference between different groups. These tests were applied in a bankruptcy prediction study using earnings management information (*Séverin & Veganzones, 2021*). The research focused on only seven features. $T$-test and K–W showed that most attributes are statistically significant at the level of 1%. The K–W method identified that the ratio of Financial Expenses and Total Assets is significant only at the 5% level. A study utilized the K–W approach to examine whether the FS methods suggest employing statistically significant features (*Papíková & Papík, 2022b*). Results indicated that the use of FS approaches in most cases improved the AUC score, and a statistical significance of less than 5% was rarely observed. *Ben Jabeur & Serret (2023)* focused on the usage of the $T$-test. Features having statistically significant coefficients at the 1% level were selected for further research. Experiments with different ML methods revealed that the improvement of the AUC score was only observed for the $T$-test in combination with SVM or PLS-DA ML methods; the achieved improvement was about 3%.

The feature selection based on wrapper methods depends on a particular machine learning algorithm (*Verma, 2020*). It utilizes a greedy search methodology, examining all possible feature combinations. The suitability of different feature sets is based on a performance metric that changes depending on the problem type. For instance, classification tasks use accuracy, precision, recall, F1-score, *etc*. The most popular wrapper methods are backward/forward stepwise logistic regression (stepwise LR), stepwise discriminant analysis (stepwise DA) and generic algorithm (GA). An unspecified backward wrapper method was applied to the Italian SME's data (*Perboli & Arabnezhad, 2021*). Based on the assumption that important variables should increase the accuracy by at least 1%, the method identified a subset of 15 important features out of 170 possible.

A fairly new Population Diversity Controller-Genetic Algorithm (PDC-GA), which is an improved version of the simple GA FS method, combining GA with k-mean clustering, was used to forecast the bankruptcy of Polish enterprises (*Al-Milli, Hudaib & Obeid, 2021*). The results indicate that the new FS algorithm improves the results of the GA FS method from 1% to 4%. After analyzing a year's worth of data, the PDC-GA approach, when paired with the KNN ML approach, yielded a 0.9573 AUC score, whereas the GA FS approach yielded a 0.9391 AUC score. Furthermore, the utilization of the GA method in the *Tsai et al. (2021)* study was unsuccessful due to its ineffective performance in comparison with

other dimensionality reduction methods (PCA, *t*-test), *i.e.,* GA resulted in an AUC score of 0.641.

Two wrapper methods, *i.e.,* stepwise LR and stepwise DA, were used to forecast French enterprises' bankruptcy (*Ben Jabeur, Stef & Carmona, 2022*). Both wrapper methods selected less than 10 features due to the high correlation between features. The results indicated that stepwise LR and stepwise DA achieved similar results, but the XGBoost feature selection technique achieved a higher AUC score of 0.958, while the highest AUC for stepwise LR and stepwise DA was 0.935 and 0.940, respectively. In *Papíková & Papík (2022a)*, the highest AUC value (99.95%) was achieved using the CatBoost classification method for random oversampling with a wrapper (stepwise regression) feature selection method.

Embedded methods combine the advantages of filter and wrapper methods. These methods perform feature selection and training of the algorithm in parallel. One of the embedded techniques used in financial distress or bankruptcy feature selection is the least absolute shrinkage and selection operator (LASSO). This technique eliminates uninformative predictors from the model by shrinking their coefficients to zero (*Altman et al., 2022*). Research using the LASSO technique can be divided into two categories: (1) studies that apply LASSO as a step in data preparation, and (2) studies that use LASSO for comparing dimensionality reduction methods. *Huang, Wang & Kochenberger (2017)* and *Li et al. (2021)* belong to the first category. In this case, the training sample consisted of up to 200 companies described by 90 features. However, the final models included only seven–eight features. Research analyzing 2,040 SMEs from Croatia using the LASSO algorithm selected 21 features out of 87 possible (*Altman et al., 2022*). Moreover, the LASSO technique could be used as one of the FS voting classifier techniques (*Du et al., 2020*). Studies belonging to the second category (*Jiang et al., 2021*; *Papíková & Papík, 2022a*; *Volkov, Benoit & Van den Poel, 2017*) show that the LASSO technique efficiency is lower compared with other FS techniques such as stepwise regression or random forest. However, the results obtained in *Papíková & Papík (2022a)* can be considered controversial because the authors claim that the impact of individual feature selection methods has no statistically significant effect on the models' performance. Moreover, in this case, the total feature space consisted only of 27 features. An interesting analysis perspective is the stability of the LASSO method over the years (*Zizi et al., 2021*); the first year, LASSO identified seven important features, the next year –9, and only four of them remained the same.

Another embedded technique often used for financial distress identification is XGBoost. *Ben Jabeur, Stef & Carmona (2022)* and *Du et al. (2020)*, using XGBoost method, identified nine–10 features, which cover 10–20% of the entire feature set. In comparison, tree-based, L1-based and f–classif methods selected 20–30% of the entire feature set (*Du et al., 2020*). However, the impact on modelling accuracy was not compared. The final model was developed using a single combined set of features. The CatBoost boosting technique was used for feature selection (*Jabeur et al., 2021*). CatBoost automatically ranks the most important features and uses them for forecasting. The results show that CatBoost, with its feature importance ranking, performs well as it achieves an AUC score of 0.994. However, this study did not consider other FS techniques.

Logistic regression analysis (LRA) is an embedded feature selection technique used in the financial distress prediction study (*Doğan, Koçak & Atan, 2022*). After applying the feature selection method, the cross-validation rate increased from 87.21% to 90.06%. However, the results of the study should be interpreted with caution because the data set is small, consisting of 172 stock market enterprises, of which 71 experienced financial difficulties. Another small data set (266 enterprises with 17 financial ratios) was used for French enterprises' bankruptcy prediction (*Ben Jabeur & Serret, 2023*). This study used a Partial Least Squares Discriminant Analysis (PLS-DA) FS approach. The results were assessed over a period of one year by employing diverse classification ML techniques. The proposed feature selection method improved the AUC score results of SVM and PLS-DA ML models about 2%. After the year, the authors expanded their research by including additional FS methods, namely XGBoost, stepwise LR, stepwise DA and PLS-DA (*Ben Jabeur, Stef & Carmona, 2022*). In this study, the PLS-DA method achieved similar results to the wrapper methods but was inferior to the XGBoost FS method, scoring 0.3%–2.7% lower AUC.

The Filter and wrapper FS approach and their hybrid methods were examined in a study focusing on the financial distress in Taiwanese electronics industry businesses (*Lin & Hsu, 2017*). The hybrid FS method achieved 87.77% accuracy, the filter FS method –75.10%, and the wrapper –82.33%. The authors claim that this hybrid FS method keeps wrapper approaches' advantages while saving computational resources.

In conclusion, the authors use feature selection techniques for the reduction of the main dimensionality problems, such as data sparsity, multiple testing, and overfitting (*Kuiziniene et al., 2022*). Moreover, based on the analysis of feature selection, three main directions of research can be distinguished: (1) research that uses one feature selection method as a data preprocessing step; (2) research that uses several feature selection methods as a data preprocessing step; (3) research that compares and analyzes several feature selection methods. The third direction proved the effectiveness of CFS (*Faris et al., 2020*), stepwise LR for a one-year period, $t$-test for a two-year period (*Ben Jabeur & Serret, 2023*), XGBoost (*Ben Jabeur, Stef & Carmona, 2022*), and FS voting classifier (*Du et al., 2020*) methods.

## Feature extraction

The feature extraction (FE) approach is a process of transforming high-dimensional data into a lower-dimensional feature space (*de Freitas & de Freitas, 2013*). This approach is frequently used for visual information representation (*Kuiziniene et al., 2022*; *Ye, Ji & Sun, 2013*). For instance, multidimensional scaling MDS (*Khoja, Chipulu & Jayasekera, 2019*; *Mokrišová & Horváthová, 2020*) and tSNE (*Zoričák et al., 2020*) approaches were used for data visualization in the context of financial distress and bankruptcy. A comparative study of different feature extraction methods, including linear discriminant analysis (LDA), principal component analysis (PCA), locality preserving projections (LPP), neighborhood preserving embedding (NPE), and robust subspace learning (RSL) were presented in *Wang, Liu & Pu (2019)*. Feature extraction approaches include linear and non–linear methods.

Linear methods reduce dimensionality by using linear functions (*Ayesha, Hanif & Talib, 2020*). For example, principal component analysis (PCA) is a projection-based method

in which the main assumption is that the behavior of data could be represented by a significant variance (*Sulistiani, Widodo & Nugraheni, 2022*). It is noted that authors tend to select reduced dimensions in three manners: (1) experimentally(by selecting several reduced dimensions) (*Adisa et al., 2019*); (2) based on a pre-defined covariance ratio threshold (usually in the range of 60%–90% of the original information) (*Jiang et al., 2021*); (3) hybrid (numerical determination by experimentally choosing a percentage of the explained variance) (*Sulistiani, Widodo & Nugraheni, 2022*). Moreover, PCA is usually followed by a balancing technique. A more interesting application of PCA is discussed in *Jiang et al. (2021)*. In this case, PCA was used to reduce the dimension of word vectors generated by BERT (Bidirectional Encoder Representations from Transformers). Another interesting study (*Acharjya & Rathi, 2021*) conducted through the use of PCA identified 15 attributes as the main features out of the total of 64 conditional attributes.

The main advantage of the non–linear methods is a better representation of the real-world data, which is more often non–linear than linear (*van der Maaten, Postma & Herik, 2007*). Kohonen's self-organizing map (SOM) is an example of such methods being used in the financial distress domain (*Mora García et al., 2008*). However, in this research, the SOM technique was applied before splitting the data into training and testing sets and thus violating the requirement of unseen data. Another non–linear method often used in bankruptcy prediction is the autoencoder. This technique reduced the dimensionality of the data from 64 to 16 features (*Soui et al., 2020*). Moreover, the stacked autoencoder with softmax classification function achieved the highest AUC (87.9%) score, compared to other methods (SVM, RF, XGBoost, *etc.*).

The comparison of feature selection and extraction methods provided in Table 1 shows that comprehensive and high-performance feature extraction is inseparable from high computational costs.

## DATA

The dataset used in this study was collected and provided by LTD ''Baltfakta''. It contains information on 81,202 active enterprises operating in Lithuania, covering the period from 01–01–2015 to 30–05–2022. The analyzed enterprises meet the following conditions:

1. legal status belongs to one of the following categories: (a) a private limited liability; (b) a public limited liability; (c) an individual enterprise, (d) a small community. The frequency of the number of enterprises in the market has been considered when choosing the statuses. Moreover, all these types of enterprises share the objective of increasing it's worth;
2. does not belong to the following Nace sectors: K – financial and insurance, L – real estate, O – public administration and defense, compulsory social security, these sectors have been eliminated because of different accounting requirements for financial statements;
3. enterprise's age is ≥ 1.5 years in order to reduce the number of missing values in the data;
4. enterprise has provided at least one financial statement from the last two years;

5. enterprise has one or more social insured employee, avoiding inactive market participants;

6. at least 1.5 years have passed since good status recovery. Only enterprises having additional registration of good enterprise conditions (*e.g.*, the enterprise had a history of bankruptcy case in court, but after a change in circumstances, the enterprise's activities continued, and its good conditions were registered in the Lithuanian register) were analyzed. This criterion has been incorporated to re-evaluate enterprises that have experienced financial difficulties in the past, yet persist in their operations.

## Class

In this study, we analyze a binary classification problem, where 0 indicates "good" enterprise conditions and 1 stands for "bad" enterprise conditions. We define a "bad" situation as financial distress that is detected at least in one Lithuanian government register and meets the following conditions:

1. a bankruptcy case is filed against the enterprise;
2. the enterprise's status changed to going bankrupt, bankrupt, under restructuring, under liquidation, liquidation being initiated, removed, liquidated, liquidation due to bankruptcy;
3. the enterprise has made announcements to the register center about bankruptcy, liquidation, restructuring, insolvency, *etc.*;
4. the enterprise is included in the State tax inspectorate's lists of (1) companies temporarily exempted from submitting declarations to the STI, (2) companies that have declared temporary inactivity of the STI; (3) companies for which the STI has submitted a proposal to the State Register for deregistration in accordance with Article 2.70 of the Civil Code.
5. enterprise had no socially insured employees for the past six months.

The final dataset consists of 274,105 unique records, of which only 2.88% cases represent financial distress.

## Features

The data can be divided into ten different categories, depending on the provider of data or its held information (see Table 2). For example, three data source providers are combined in the *Sector's* category: (1) sectors type identified by NACE category; (2) information on sector profitability, competitiveness, *etc.* from the Lithuanian Statistics Department; (3) sectoral indicators calculated by combining financial statement data and NACE types. The "No." column indicates how many features fall under each category. The data type indicates whether the feature is a primary data feature or an additional feature derived from primary data. For instance, company assets or the number of employees in December, are assigned to the primary data type, whereas ROA, gross profit, change of employees per year, *etc.*, belong to the additional data. The data frequency is divided into three categories:

1. stable: information is constant, *e.g.*, legal status, types of sectors;
2. depending on an event: changes when the event occurs, the number of courts, the number of changes of directors, the time elapsed since the last event, *etc.*;

**Table 2  Characteristics of the features.**

| # | Data category | No. | Data type | | | | Periodicity | | |
|---|---|---|---|---|---|---|---|---|---|
| | | | Primary | Additional | Stable | Depending on event (DoE) | Annual | Quarterly | Monthly |
| 1. | Board, top managment shareholders | 21 | | x | | x | | | |
| 2. | Financial statements | 160 | x | x | | | x | | |
| 3. | Enterprise type | 4 | x | | x | | | | |
| 4. | Legal events | 13 | x | x | | x | | | |
| 5. | Macro | 247 | x | x | | | x | x | x |
| 6. | Sector's | 121 | x | x | x | | x | | |
| 7. | Social Insurance | 391 | x | x | | | x | | x |
| 8. | Taxes | 5 | x | x | | | x | | |
| 9. | Other | 10 | x | x | | x | | | |
| | **Total:** | **972** | | | | | | | |

[1] This study appendices are held in a GitHub repository: https://github.com/DovileKuiziniene/FS_and_FE_for_FD_Appendices.

3.  periodic data (annual, quarterly, monthly): information that is updated at the indicated periodicity, *e.g.*, financial reports, macro indicators, the number of employees.

All categorical features have been transformed into binary features by expanding the feature space. In the final data set, each enterprise is described by 972 features for each analyzed year. The list of all features is presented in different tables by data category (see Table 2) and shown in Appendix A: Table A.1, A.2, A.3, A.4, A.5, A.6, A.7, A.8, A.9, A.10, A.11, A.12, A.13, A.14, A.15[1]. However, not all features are included in the analysis due to not having the variance condition. These features in Appendix A are crossed out and eliminated from the study.

The data contains 972 features, of which 951 are continuous and 21 binary. Categorical features describe the enterprise type and define belonging to one of the 17 possible sectors (based on the Nace classifier). Most enterprises (∼87%) are private limited liability enterprises, ∼12% of companies are small communities, and only ∼1% of data consists of public limited liability and individual enterprises. The distribution of Nace values shows that most of the enterprises belong to the Nace C–manufacturing (10.2%), Nace F–construction (11.35%), Nace G– wholesale and retail trade; repair of motor vehicles and motorcycles (28.61%), Nace H–transportation and storage (10.54%), Nace M–professional, scientific and technical activities (12.1%).

Figure 1 represents a distribution of feature values according to classes, *i.e.,* financially distressed and healthy enterprises. Each figure shows one of the most significant features in each data category (see Table 2). In this case, significance is determined using a vote-based feature selection covering all feature selection methods (see 'Dimensionality reduction methods'). Moreover, feature names contain information regarding the time perspective, *i.e.,* Y indicates the number of years before the outcome; Q represents a quarter of the analyzed year; M shows a month of the analyzed year. If there is no time indication, the feature is affected by the occurrence of certain events. For example, a director change would affect variables representing the frequency of this event and the time since the last change. The distribution of the most significant features justifies the presence of an

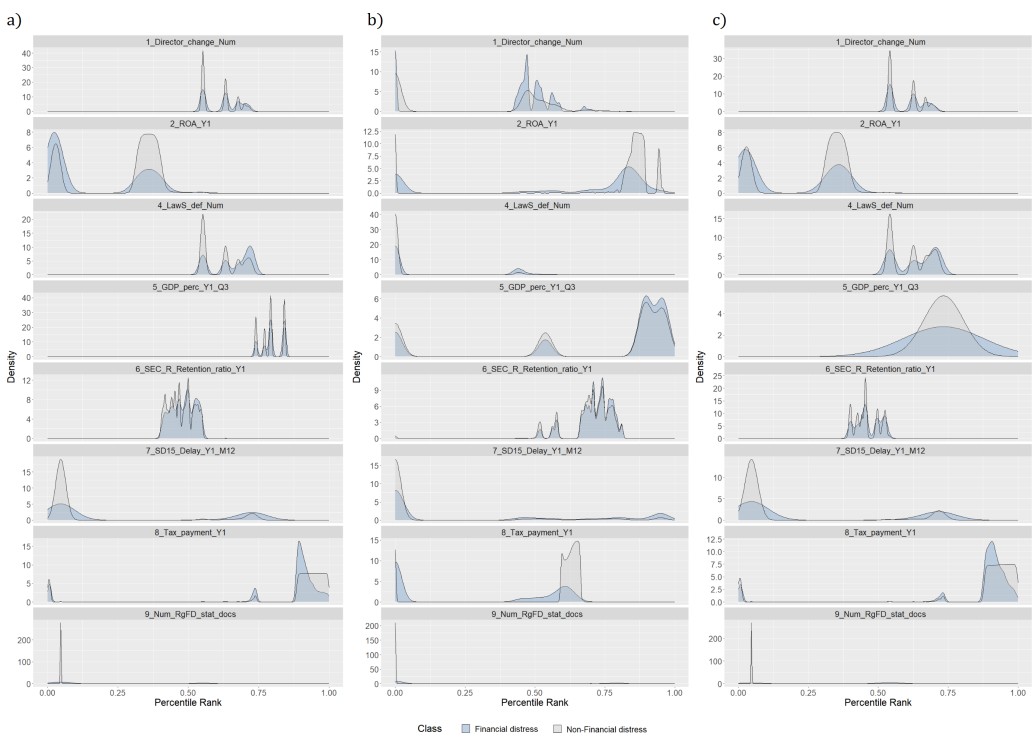

**Figure 1** (A–C) Distribution of features by classes.

invisible "optimal cut-off point" among financial distress and healthy enterprises, discussed in *Beaver (1966)*. Hence, more advanced artificial intelligence (AI) techniques should be used to identify the signs of financial distress.

Density plots are rescaled histograms, which use kernel density estimation to reveal the probability density function (PDF) ($Y$-axis) of the specific feature (*Wilke, 2023*). A better understanding of the data comes from comparing density plots for different data sets. For example, density plots of features like the history of bad events (9_*Num_RgFD_stat_docs*) for training and testing data sets are almost the same. This tendency is advantageous due to the ML patterns learning process in the training set, which is also reflected in the test sample. The same tendency could be seen in features 8_*Tax_payment_Y*1, 7_*SD*15_*Delay_Y*1_*M*12, 6_*SEC_R_Retention_ratio_Y*1 and 2_*ROA_Y*1. Nevertheless, upon examination of the 1_*Director_change_Num* and 4_*LawS_def_Num* variables, it is observed that the training component is divided into two distinct segments of the trend, of which only one remains in the testing sample. Furthermore, the values of the trend in the training and testing samples for the 5_*GDP_perc_Y*1_*Q*3 variable differ substantially.

## METHODOLOGY

The study aims to identify financial distress using an optimal number of informative features. For this reason, we employ different dimensionality reduction methods in

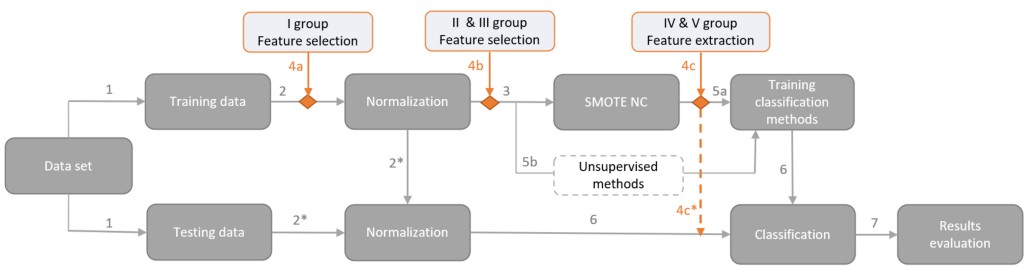

**Figure 2  Proposed framework for financial distress identification.**

combination with various machine learning techniques. Our work is motivated by the following research questions:

RQ1: Which feature selection or extraction method is the most efficient for financial distress identification?

RQ2: Which data categories are the most important in financial distress prediction?

RQ3: Does the composition of the final feature set depend on the feature selection technique?

RQ4: Which feature extraction strategy is more efficient: analyzing all features at once *vs* analyzing features by categories?

RQ5: Which machine learning model performs better in financial distress identification?

The proposed framework for financial distress identification is presented in Fig. 2. The first step deals with data overfitting by splitting the data into training and testing sets. Cross–validation is not used in order to have testing results on the latest data, which will give market participants as relevant information about model performance as possible. Hence, the testing data set covers the latest period, *i.e.,* the class variable is based on the first half of the 2022 year, and the training set consists of class identifiers covering the period from 2018 to 2021. Thus, a data set is divided into training and testing sets according to a ratio of 74:26. However, the usage of other data preparation steps depends on the used dimensionality reduction method (the fourth arrow), *i.e.,* it can come before or after dimensionality reduction method implementation (see. Figure 2).

The second step is the process of data normalization. Normalization is an important step for designing classification models aiming to obtain comparable scales of criteria values. First, we normalize the training set and then perform normalization on a testing set using the same normalization characteristics (identified by 2⋆ arrow in Fig. 2). Therefore, the training data is normalized immediately without regard to the year, and the test data for 2022 is normalized based on the normalization estimates of the training data. Before normalization, several values of financial ratios were classified as outliers due to unrealistic values of the feature. These values were changed to NA value (not available data). We use Min-max normalization to scale the variables between zero and one (*Xu, Fu & Pan, 2019*):

$$x' = \frac{x - x_{min}}{x_{max} - x_{min}},$$
(1)

where $x$ is an original value of the feature, $x'$ – transformed value of a feature, $x_{min}$ and $x_{max}$ are respectively minimum and maximum values of a feature.

After normalization, all missing values (NA) are replaced by the smallest value – zero.

The third step is data balancing, which can significantly impact the performance of classification methods. The lack of representation from minority classes in the data sets hinders the discovery of underlying patterns (*Lin et al., 2017*; *Fernández et al., 2018*). To balance the training set, the SMOTE–NC approach (*Wongvorachan, He & Bulut, 2023*) was used, which is suitable for data containing continuous and categorical variables. After balancing, the training data has 392,734 unique records.

The fourth step represents the usage of dimensionality reduction methods. The main research steps (indicated by the dark gray color) remain constant, whereas the dimensionality reduction methods (highlighted in orange) are applied only once, *e.g.*, if a method from the I group is used (4a in Fig. 2), then methods from other groups (4b or 4c in Fig. 2) are not involved (each group and the methods assigned to it are presented in the Section 'Dimensionality reduction methods'). Moreover, the groups of dimensionality reduction methods must be placed at these precise places due to the specificity of the selected modelling techniques. For instance, Kruskal–Wallis or Cohen's D tests should be applied before data normalization (4a in Fig. 2); if PCA (4c in Fig. 2) is performed before SMOTE–NC, the extracted structure would be distorted and the balancing technique would introduce additional noise. Feature extraction for the testing set (*i.e.,* IV and V groups; 4c in Fig. 2) is implemented in the same manner as normalization, *i.e.,* the test set is projected onto the reduced feature space, obtained during the training.

The fifth step is model training, including supervised (5a in Fig. 2) and unsupervised (5b in Fig. 2) methods. Supervised ML methods require the implementation of data balancing techniques to create an efficient model, capable of recognizing both classes. Therefore, the SMOTE–NC technique was used, which creates data in a continuous or categorical way, depending on the feature type. However, unsupervised ML (5b in Fig. 2) methods do not require the balancing of the training set due to a different approach to the problem. Using unsupervised methods, a minority class is identified as an anomaly or an outlier. Hence, the number of such classes is sufficient and there is no need to additionally increase the number of minority classes (identified by step 5b in Fig. 2). Nevertheless, the use of the feature extraction approach (IV and V groups) requires the use of SMOTE–NC for all experiments, to ensure that the initial space and outcomes are identical. After feature extraction, the added rows from SMOTE–NC are removed for unsupervised ML methods (see 4c* arrow in Fig. 2).

The sixth and seventh steps represent the testing set classification and evaluation of the results.

## Dimensionality reduction methods

A large number of features can cause problems related to data sparsity, multiple testing, multicollinearity and overfitting (*Kuiziniene et al., 2022*). Different dimensionality reduction techniques have been developed to overcome these problems (see Fig. 3). In this study, several dimensionality reduction techniques have been selected based on their

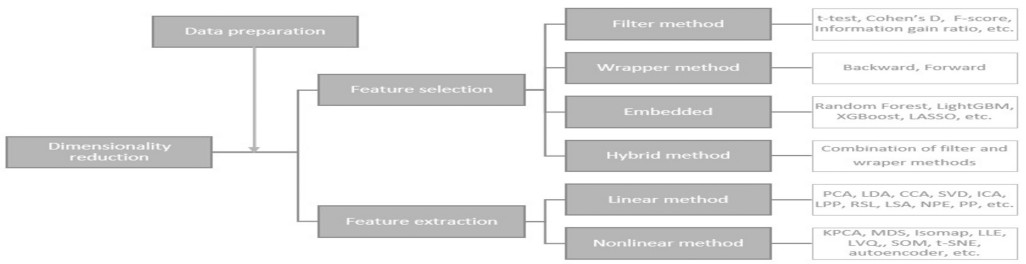

**Figure 3   Dimensionality reduction methods (*Kuiziniene et al., 2022*).**

relevance concerning different data types (continuous and categorical features), running time, overfitting, and suitability to unseen data.

These dimension reduction techniques can be classified into several categories:

1.  Feature selection approach (FS) is used to determine a small subset of informative features from the original large range of data (*Al-Tashi et al., 2020*) by removing irrelevant, redundant or noisy features. Let $Y = \{y_1, y_2, \ldots, y_n\}$ denote the class labels, where $n$ is the number of enterprises and $y_i \in \{0,1\}, i = 1, \ldots, n$ belongs to one of the two classes $c_1$ – non-financial distress or $c_2$ - financial distress. Each enterprise is defined by a number of features $x_{ij}, j = 1, \ldots, d$. We assume that $x_{ij}$ are realizations of features $X_j$ that can be ordered according to their importance

    $$FI = (X_{\pi(1)}, X_{\pi(2)}, \ldots, X_{\pi(d)}) \qquad (2)$$

    where $\pi(j), j = 1, 2, \ldots, d$ is the new descending index of feature $X_j$.

    (a)  Group—Filter methods assume complete independence between the model and the data. In this case, features are selected based on their scores in various statistical tests, determining their correlation with the outcome variable.

        i.  Cohen's D *(ChD)* coefficient is a standardized effect size for comparing the means of *Almasri (2022)*; *Imik Tanyildizi & Tanyildizi (2022)* two groups (financial distress enterprises and non-financial distressed). Cohen's D is defined as

        $$ChD = \left| \frac{\overline{X}_{x \in c_1 j} - \overline{X}_{x \in c_2 j}}{\sqrt{\frac{\sigma^2_{x \in c_1 j} + \sigma^2_{x \in c_2 j}}{2}}} \right| \qquad (3)$$

        where $\overline{X}_{*j}$ and $\sigma_{*j}$ are means and standard deviations of corresponding groups. The order of the feature's importance is determined by the effect size, *i.e.*, the greater the *ChD*, the larger the effect size.

        ii.  Correlation *(Corr)* coefficient analyzes possible linear association between variables (*Cheng et al., 2020*; *Jiarpakdee, Tantithamthavorn & Treude, 2020*; *Séverin & Veganzones, 2021*). Based on the type of feature, we use Pearson and point-biserial correlation:

        $$Pearson\ correlation = r_{X_j X_k} = \left| \frac{\sum_{i=1}^{n}(x_{ij} - \overline{X}_j)(x_{ik} - \overline{X}_k)}{\sigma_{X_j} \sigma_{X_k}} \right| \qquad (4)$$

$$Point-biserial\ correlation = r_{pb} = \left| \frac{\overline{X}_{x \in c_1 j} - \overline{X}_{x \in c_2 j}}{\sigma_{X_j}} \sqrt{pb} \right| \qquad (5)$$

where $p$ and $b$ denotes the proportion of cases in the classes of â financial distressâù and "non-financial distress". $\sigma_{X_j}$ is a standard deviation of feature $j$ for all enterprises. Correlation with the outcome variable determines the order of the feature's relevance (higher is preferable). A high correlation ($> 0.75$) between features indicates the problem of multicollinearity. To deal with this problem, from the group of highly correlated features, only one is left, having the highest correlation with the outcome, while others are removed from the sample. Finally, all the remaining features are ranked by the strongest relationship with the outcome.

iii. The Kruskal–Wallis (K–W) test is used to compare medians among two or more groups. The Kruskal–Wallis test is chosen because it does not require any prior assumptions about data distribution and is good for all types of features (*Devi Priya et al., 2022*). The selected significance level is 0.05, *i.e.,* smaller $p$ values suggest that there are substantial differences between the two groups (*Linja et al., 2023*). The test statistic for the Kruskal–Wallis test is defined as follows:

$$Kruskal-Wallis\ \chi^2 = H = (n-1)\frac{\sum_{l=1}^{c} n_l (\overline{R}_l - \overline{R})^2}{\sum_{l=1}^{c} \sum_{j=1}^{n_l} (R_{lj} - \overline{R})^2} \qquad (6)$$

where $n$ – the sample size, $n_l$ is the number of observations in class $l$, $R_{lj}$ is the rank of feature $j$ from class $l$, $\overline{R}_l$ the mean rank of all features in class $l$, $\overline{R}_l$ is the mean of all the $R_{lj}$, $\overline{R}$ is the mean of all the $R_{lj}$. The order of feature importance can be defined by the lowest $p-value$ or the highest $|H|$ value.

(b) Group—Embedded methods include the feature selection in the model fitting process. Classification methods that perform embedded feature selection include:

i. The least absolute shrinkage and selection operator (LASSO) is a method that combines feature selection and regularization (*Altman et al., 2022*; *Zizi et al., 2021*; *Huang, Wang & Kochenberger, 2017*). The LASSO has the advantage of maintaining the stability of ridge regression and outperforming stepwise regression models (*Li et al., 2021*). The LASSO function aims to solve (*Tibshirani, 1996*):

$$\min_{\beta_0, \beta_i} = \left\{ \sum_{i=1}^{n} (y_i - \beta_0 - x_{ij}^T \beta)^2 \right\} \qquad \text{s.t.} \quad \sum_{j=1}^{d} \left| \beta_j \right| \leq \lambda. \qquad (7)$$

where $\beta := (\beta_1, \beta_2, \ldots, \beta_d)$ is a coefficient vector and $\lambda > 0$ adjusts the sparsity of the estimator by eliminating features from the model forcing $\beta_j \to 0$. The importance of a feature is the absolute value of the LASSO regression coefficient.

ii. Random Forest (RF) uses a permutation importance measure, which is shown to be an effective tool for feature selection (*Gregorutti, Michel & Saint-Pierre, 2015*). RF offers two techniques for calculating the relevance of all features, resulting in a feature importance rank. These techniques create a rank by

considering both the feature cost and differentiating ability (*Zhou, Zhou & Li, 2016*):

A. Mean Decreasing accuracy (MDA) (*Han, Guo & Yu, 2016*) is a feature importance measure based on *OOB* (out of bag) error. Assume $h_t(X_i)$ and $h_t(x_{ij})$ refer to the predicted label for OOB instance $X_i$ before and after feature permutation (*Wang, Yang & Luo, 2016*). The relevance of a feature $X_j$ is assessed by *MDA*, which estimates the mean drop in *OOB* accuracy before and after feature $X_j$ permutation:

$$MDA(X_j) = \frac{1}{n_{tree}} \sum_{t=1}^{n_{tree}} \frac{\sum_{i \in OOB} I(y_i = h_t(X_i)) - \sum_{i \in OOB} I(y_i = h_t(x_{ij}))}{|OOB|} \qquad (8)$$

B. Mean Decreasing Gini (MDG) calculates the overall reduction in node impurity (*e.g.*, Gini index) as a result of splitting on the feature and averaging it across all trees (*Wang, Yang & Luo, 2016*). The MDG is defined as follows:

$$MDG(X_j) = \frac{1}{n_{dot}} \left[ 1 - \sum_{j=1}^{n_{dot}} Gini(l)_j \right] \qquad (9)$$

where $Gini(l)_j$ is the $j$th Gini index of feature $X_j$ among the $n_{dot}$ tree nodes. The Gini index is defined as follows:

$$Gini = \sum_{l=1}^{c} l_j(1 - l_j) = 1 - \sum_{jl=1}^{c} l_k^2 \qquad (10)$$

Where $l_j$ is the probability of the feature $X_j$ classification in the distinct class (*Zhang et al., 2019*).

iii. Extreme Gradient Boosting Machine (XGBoost or XGB) method assigns a relevance value to each feature based on its involvement in the outcome decision-making using boosted decision trees by *Gain* metrics (*Zheng, Yuan & Chen, 2017*) ((11)).

$$Gain = \frac{1}{2} \left[ \frac{(\sum_{i \in I_L} g_i)^2}{\sum_{i \in I_L} h_i + \gamma_1} + \frac{(\sum_{i \in I_R} g_i)^2}{\sum_{i \in I_R} h_i + \gamma_1} + \frac{(\sum_{i \in I} g_i)^2}{\sum_{i \in I} h_i + \gamma_1} \right] - \gamma_2 \qquad (11)$$

where $g_i$, $h_i$ indicate the first and second-order gradients, $\gamma_1$ and $\gamma_2$ are regularization parameters, $I = I_L \cup I_R$, while $I_L$ and $I_R$ represent all the left and all the right nodes after each split, respectively (*Jiang et al., 2023*). The greater the XGBoost gain, the more useful and significant the feature (*Chen et al., 2021*).

(c) Group—Hybrid combines different methods together:

i. Overlapping Features (*Over_feat*)–features having a rating in all feature selection methods, *e.g.*, if LASSO excludes a feature it will not be further analyzed.

ii. Voted importance (*Voted_imp*)–a joint rank combining results from all feature selection methods. Each feature selection method ranks the features from the most important to the least. Based on these ranks, we can calculate the

cumulative weight for each feature:

$$Voted\_imp(X_j) = \sum_{j=1}^{m} FI_j \tag{12}$$

where $FI_j$ is the rank of the feature and $m$ is the number of used feature selection approaches (in I and II group).

2. Feature extraction approach (FE) maps high-dimensional data $X$ into a new, lower-dimensional space ($Z = [z_1, z_2, \ldots, z_n] \in \mathbb{R}^{n \times k}$, where $k < d$) retaining as much information as possible (*Ayesha, Hanif & Talib, 2020*):

(a) Group—Linear methods a transformation of data, which projects data linearly:

i. Linear Discriminant Analysis *(LDA)* projects the data into a new feature space by minimizing the scatter within classes ($S_w$) and maximizing the scatter between classes($S_b$) (*Djoufack Nkengfack et al., 2021*). This method only creates as many linear discriminants as there are number of classes, minus one (*Anowar, Sadaoui & Selim, 2021*).

$$S_w = \frac{1}{n} \sum_{l=1}^{c} \sum_{i=1}^{n_l} (X_{il} - \overline{X}_g)^T (X_{il} - \overline{X}_l) \tag{13}$$

$$S_b = \frac{1}{n} \sum_{l=1}^{c} n_l (\overline{X}_l - \overline{X})^T (\overline{X}_l - \overline{X}) \tag{14}$$

where $c$ is the number of classes, $\overline{X}$ – the global mean of all classes (*Anowar, Sadaoui & Selim, 2021*; *Djoufack Nkengfack et al., 2021*), and eigenvectors are defined as:

$$W = eig(S_w^{-1} S_b) \tag{15}$$

where the largest eigenvalues correspond to $k$-eigenvectors of ($S_w^{-1} S_b$), and of linear discriminants forms $W = (w_1, w_2, \ldots, w_k)$ (*Djoufack Nkengfack et al., 2021*; *Ayesha, Hanif & Talib, 2020*)

$$Z^* = XW^T \tag{16}$$

ii. Principal component analysis (PCA) is the most extensively used dimensionality reduction approach summarizing a large set of features in low dimension with minimum loss of information (*Ma & Park, 2022*). The PCA method transforms a set of features into uncorrelated Principal Components (PCs), which are ranked and ordered by their importance, *e.g.*, the first PC explains the most variance, the second PC explains the most variance in what is left once the effect of the first PC is removed, *etc.* (*Ayesha, Hanif & Talib, 2020*). The general form of PCA transformation can be estimated as a linear weighted combination of features:

$$Z^* = XW \tag{17}$$

where $Z = [z_1, z_2, \ldots, z_n] \in \mathbb{R}^{n \times d}$ are principal components and $W = [w_1, w_2, \ldots, w_d] \in \mathbb{R}^{d \times d}$ are eigenvectors by *Ayesha, Hanif & Talib (2020)*. The first eigenvector is determined as follows:

$$w_1 = \underset{||w||=1}{\arg\max} \left\{ ||Xw||^2 \right\} \tag{18}$$

iii. The factorial analysis of mixed data (FAMD) was proposed for handling data containing continuous and categorical features (*Ran, 2019*). FAMD is a combination of PCA and MCA (multiple correspondence analysis) methods (*Ma & Park, 2022*). Based on the types of features, relationships between them are measured using Pearson's correlation coefficient, the squared correlation ratio or the chi-squared test (*Momenzadeh et al., 2021*; *Josse, 2016*).

$$F_s = \underset{F_s \in \mathbb{R}^n}{\text{argmax}} \sum_{j_{cont}=1}^{p_{cont}} r^2(F_1, X_{j_{cont}}) + \sum_{j_{cat}=1}^{p_{cat}} \eta^2(F_1, X_{j_{cat}}) \tag{19}$$

where $F_s$ is the orthogonal condition to $F_s'$ for all $s' < s$, $p_{cont}$ and $p_{cat}$ are the number of continues and categorical features, respectively (*Josse, 2016*). The general transformation of FAMD can be expressed as:

$$Z' = XF_s \tag{20}$$

iv. Union of separate PCA & FAMD models (*Union_PCA_FAMD*) select the appropriate data transformation depending on a data type in the analyzed category (see Table 2). In total, there were 12 different categories analyzed, and for three of them, subcategories were created: (1) *Financial statements*: records and ratios; (2) *Sector*: class and ratios; (3) *Social insurance*: employees and delay of taxes (including analyzes of time and amount). The PCA or FAMD method is chosen depending on the data type (continuous or categorical).

$$f(x) = \begin{cases} Z^*, & \text{if all features belonging to the category are continuous} \\ Z', & \text{if category contains mixed type features} \end{cases} \tag{21}$$

(b) Group – Non–linear methods reduce the dimensionality by reflecting intrinsic non–linear relations between variables.

i. Autoencoder (AE) is a neural network which is very efficient in generating abstract features of high–dimensional data by minimizing reconstruction loss (*Meng et al., 2017*). This loss function sets the target value equal to the input by applying a backpropagation algorithm. The main idea of the process is the usage of the encoder–decoder. Firstly, the encoder process transforms input data to lower-dimensional space (to a bottleneck or middle layer); secondly, the decoder converts it back into high-dimensional space (*Yan & Han, 2018*). The minimization of reconstruction loss assures that the middle layer incorporates most of the information from the original input space (*Phadikar, Sinha & Ghosh, 2023*). Thus, the middle layer holds the reduced representation of the input data (*Kunang et al., 2018*).

$$X' = g_\theta(Z) = s(W'Z + b_Z) \tag{22}$$

where $f(X)$ is an encoder function, $g(Z)$ is a decoder, $s$ is an activation function, $W$ represents weights and $b$ is the bias for data $x$. The training process of the autoencoder finds parameters $\theta = (W, b_X, b_Z)$ by minimizing the reconstruction loss (*Kunang et al., 2018*):

$$\theta = \min L(X, X') = \min L(X, g((f(X)))) \tag{23}$$

ii. *Union of separate autoencoders* (*Union_AE*) trains several autoencoders depending on the analyzed data category. Analyzed data categories are the same as in *Union_PCA_FAMD*.

## Number of features

The scientific literature lacks information on how much reduction can be performed on the original data without negatively impacting the model accuracy, *i.e.,* how to identify the minimum data set necessary for efficient modelling. In order to fill this gap, this study analyzes different subsets of features and their influence on the model's performance. For methods based on certain thresholds, a fixed set of effective features is determined, *i.e.,* LASSO algorithm $k = 24$, *Over_feat* $k = 24$, and for ChD $k = 14$. Nonetheless, there is no established methodology for determining the threshold, thus an established set of effective features has been determined experimentally by utilizing a diverse number of features from a lower-dimensional space, *i.e.,* $k \in \{15, 30, 50, 100\}$.

In the case of feature extraction, the number of features is predefined by a specific threshold, *e.g.,* for LDA $k_{LDA} = number\ of\ classes - 1$ (*Anowar, Sadaoui & Selim, 2021*), for PCA we analyze features whose cumulative variance exceeds a predefined threshold (70–90% of the total variance) (*Aggarwal, 2015*), for autoencoder the number of features depends on the minimization of reconstruction loss of the training sample (*Phadikar, Sinha & Ghosh, 2023*). However, for the comparison with feature selection approaches, the selection of $k$ is the same for PCA and FAMD methods, which explains from 89.8% (100 PC's) to 67.5% (15 PC's) of the total variance. In the analysis of *Union_PCA_FAMD* we make two data sets, which explain $\geq 70\%$ and $\geq 80\%$ of the total variance in each data category, and the number of features is $k_{Union\_PCA\_FAMD} \in \{64, 93\}$, respectively. To analyze the efficiency of non–linear methods, we develop several autoencoder structures. The different number near autoencoder – I, II, III identifies a number of hidden dense layers between the input layer and the middle layer; the number of units in the middle layer is $k_{AE} \in \{32, 64, 100\}$. Finally, the *Union_AE* uses $k_{Union\_AE} \in \{42, 74\}$. It should be noted that in the case of $k_{Union\_AE} = 74$, the obtained number of features is high. Therefore, the quantity of features is reduced until all of them satisfy th $e < 0.7$ correlation requirement, resulting in a feature set comprising of 42 features.

## Machine learning methods

Machine learning (ML) techniques cover a wide range of algorithms addressing various types of problems. There are two main groups of ML algorithms: supervised and unsupervised learners (*Algren, Fisher & Landis, 2021*).

1. Supervised machine learning techniques are used to construct predictive models. These algorithms try to model relationships and dependencies between the dependent variable and independent variables. In machine learning, dependent variables are frequently referred to as labels, targets or classes and independent variables are known as features (*Algren, Fisher & Landis, 2021*). This study employs boosting and neural network techniques:

   (a) The Boosting technique seeks to increase the prediction accuracy by combining results from multiple classifiers (*Nettleton, 2014*). In this study, we focus on

AdaBoost, Categorical Boosting (CatBoost) and Extreme Gradient Boosting Machine (XGBoost) techniques.

  i. AdaBoost—an ensemble approach that trains and deploys trees one after the other. It implements the boosting technique by combining multiple weak classifiers to build one strong classifier. Each weak classifier attempts to improve the classification of samples that were misclassified by the previous weak classifier (*Misra & Li, 2020*).

  ii. Categorical Boosting (CatBoost)—a machine learning technique that is based on gradient boosting decision trees. A powerful machine learning technique called gradient boosting can handle challenges with diverse features, noisy data, and complex dependencies. Moreover, this approach can easily handle categorical features by substituting the original categorical variables with one or more numerical values. Ordered boosting is a new technique that replaces the traditional gradient estimation methods (*Zhang, Zhao & Zheng, 2020*).

  iii. Extreme Gradient Boosting Machine (XGBoost)—a tree ensemble model that can be applied for both classification and regression problems. Its main idea is to make the target function as minimal as possible while employing the gradient descent method to produce new trees based on all previous trees. While using XGBoost to solve regression issues, incrementally new regression trees are added, and the residuals of the prior model are subsequently fitted using the newly created CART tree. The final predicted value is the sum of the outcomes of each tree (*Pan et al., 2022*). Selecting the loss function is a crucial step in setting XGBoost models. Loss functions are used for classification and regression problems; specifically, we use the "binary" loss function for binary classification (*Brownlee, 2021*).

(b) Discriminant analysis (DA)—a classification and dimension reduction technique described in 'Dimensionality reduction methods' (*Fisher, 1936*). The discriminant analysis creates linear combinations separating objects into classes, assuring that the variance within the class would be minimized and the variance between classes—maximized. These directions are known as discriminant functions, and their number is equal to the number of classes minus one (*Canizo et al., 2019*). In this study, we use two types of DA: Linear DA and Quadratic DA, which enables the non–linear separation of data (*Fisher, 1936*).

(c) Decision trees (DT)—a powerful data mining tool that's often used for feature selection and classification tasks. A decision tree may be used in data mining as a primary model-building method, or for automated feature selection (*Bunge & Judson, 2005*). Decision trees assist in identifying the most important characteristics that support precise classification by assessing the relevance of each feature at each node split (*Sugumaran, Muralidharan & Ramachandran, 2007*).

(d) K-nearest neighbor (KNN)—probably the most well-know non-parametric classification technique (*Chanal et al., 2022*). This method classifies each unlabeled observation by the majority class among its $k$-nearest neighbors in the training set

(*Chanal et al., 2022*). In this study, we perform experiments with $k = 3, 5, 7$ nearest neighbors.

(e) Logistic regression (LR)—contrary to its name, LR is a classification model rather than a regression model (*Subasi, 2020*). For binary and linear classification issues, logistic regression provides a faster and more efficient solution. This classification approach performs very well with linearly separable classes and is relatively simple to implement. For the LR method, the assumption of multicollinearity is fulfilled by removing features, highly correlated with other features. Furthermore, a significance level of *p*-value is utilized, and features with a *p*-value greater than 0.05 are also eliminated. For example, using the $K_W$ feature selection approach, we have identified 100 significant features that have been used for LG model development. However, to avoid the problem of multicollinearity, we have performed additional analysis and have removed highly correlated features. Thus, the final LG model has been developed using 20 statistically significant features.

(f) Naive Bayes (NB)—based on the Bayes theorem, the naive Bayes classifier is a member of the family of probabilistic classifiers. It is based on the idea that a feature's presence in a class is independent of any other features that may also be present in that class. For example, if an enterprise has a debt of 100,000 euros to a social insurance institution, suffers financial losses, and the turnover is consistently decreasing, it can be categorized as bankrupt. All three of these independent contributing factors are considered by a naive Bayes algorithm (*Krishnan, 2021*).

(g) Neural networks are artificial intelligence models that are designed to mimic the human brain's functions (*Casas, 2020*). We use three different neural network methods:

i. Artificial neural networks (ANN (I-III))—the model that uses calculations and mathematics to imitate human psychology. The unique architecture format used by ANN models mimics a biological nervous system. The ANN models are made up of neurons that interact in a complex, non–linear manner, just like the human brain. Weighted links connect the neurons to one another (*Malekian & Chitsaz, 2021*). The hidden structure of the neuron network has been marked I-III, which indicates hidden layers between input and dense layers. After each layer (except a dense one) is implemented, a drop-out layer is excluded.

ii. Convolutional neural network (CNN (II, III, V))—a specific kind of feed-forward neural network used in AI. The input data to CNN is displayed as multidimensional arrays. It works well for a large number of labelled data. Based on the important function played by the receptive field, it gives weights to each neuron, such that it can distinguish between the relative importance of different neurons. Three different layer types create CNN's architecture: convolution, pooling, and fully linked (*Shajun Nisha & Nagoor Meeral, 2021*). The hidden structure is indicated the same as in ANN. However, this is only an indication for used conv_1d and flatten layers; input, drop out, max-pooling and dense layer are not included in the hidden layers structure calculation.

iii. Extreme learning machine (ELM (100,150,200,300))—is essentially a single feed-forward neural network. The weights between inputs and hidden nodes are distributed at random throughout its single layer of hidden nodes. As a result, the parameters of the model can be calculated without the need for a learning process, and they remain constant throughout the training and prediction stages. On the contrary, the weights that connect hidden nodes to outputs can be learned exceedingly fast (*Garza-Ulloa, 2022*).

iv. Self-organizing map (SOM)—takes a set of input data and maps it onto neurons of a (usually) two-dimensional grid. Each neuron in the 2D grid is assigned a weight vector with the same dimensionality as that of the input vector. The weights represent the associated memory. Finding a winning neuron and modifying the weights of the winner neuron and its nearby neurons are the two fundamental steps in the SOM computational process (*Köküer et al., 2007*).

(h) Random Forest (RF)—an algorithm that uses classification and decision regression trees as the foundation for its ensemble machine learning approach (*Xia, 2020*). Using the values of the predictor variables as inputs, decision tree learning attempts to build a statistical prediction model to predict the values of the response variable(s). The model is created by establishing a value for the response variable within each of the recursively partitioned predictor variable spaces, which generates a decision tree.

(i) Support vector machine (SVM)—an approach for supervised machine learning that can be applied to both classification and regression problems. The SVM algorithm seeks a hyperplane in the data space that creates the largest minimum distance (referred to as a "margin") between the samples belonging to different classes (*Satapathy et al., 2019*). In this research, SVM kernel polynomial and radial basis functions are used.

2. Data labels are not used in unsupervised learning methods. Instead, discovering a structure within a dataset, "cleaning up" the data, or grouping data points into groups are the tasks that the algorithm is expected to complete (*Algren, Fisher & Landis, 2021*). In this study, three different unsupervised methods are employed:

(a) Isolation Forest (IF)—an unsupervised, non-parametric method for anomaly detection in multivariate data. This outlier detection method seeks to isolate anomalies from the rest of the data using an ensemble of decision trees. It constructs a forest of isolation trees, or iTrees. Partitioning data is required for creating iTrees. Furthermore, iTrees are binary trees that use recursive partitioning to isolate abnormalities. These anomalies, which are "rare and diverse", are therefore "more sensitive to isolation and hence have short path lengths" (*Chater et al., 2022*).

(b) K-means clustering (Kmeans) divides the set of samples into K clusters, each described by the centroid, *i.e.,* the mean of the samples in a cluster. The $k$-means algorithm uses an iterative approach to find the optimal cluster assignments by minimizing the sum of squared distances between observations and assigned cluster centroid (*Ashour, Guo & Hawas, 2019*).

**Table 3 Confusion matrix (*Fernández et al., 2018*).**

| | | Predicted class | | |
| --- | --- | --- | --- | --- |
| | | **Non-financial distress** | **Financial distress** | **Total** |
| Actual class | Non-financial distress | TP | FN | POS |
| | Financial distress | FP | TN | NEG |
| | Total | PPOS | PNEG | N |

(c) One-class support vector machine (One class SVM)—an approach that identifies unusual observations compared to the instances of the known class. The one-class SVM attempts to classify one class of objects and separate it from all other possible objects. The classifier can correctly classify certain objects, but the others will be classified as outliers. Thus, a one-class classifier is trained to label these observations as outliers (*Seo, 2007*).

## Evaluation metrics

Different performance metrics can be obtained based on the confusion matrix (see Table 3). In Table 3, *TP* denotes the number of true positives, *TN* is the number of true negatives, *FP* is the number of false positives, *FN* is the number of false negatives, *POS* is the number of actual positives, *PPOS* is the number of predicted positives, *NEG* is the number of actual negatives, *PNEG* is the number of predicted negatives, *N* is the number of all instances (*Fernández et al., 2018*). In this study, classes are defined as follows: positive class—non-financial distress enterprises and negative class– financial distress enterprises.

The most commonly used evaluation metrics are provided in Eqs. (24)–(30) (*Kuiziniene et al., 2022*). Different accuracy measures were used to get insights into different aspects of model performance. The interpretation of the results is common for all metrics – higher values indicate better performance.

Precision—the ratio of true positives (*TP*) to predicted positives (*PPOS*).

$$Precision = \frac{TP}{TP + FP}. \tag{24}$$

Recall—the ratio of the true positives (*TP*) to actual positives (*POS*), also known as sensitivity or *TPR* (true positive ratio).

$$Recall = \frac{TP}{TP + FN}. \tag{25}$$

Specificity—the ratio of the true negatives (*TN*) to the actual negatives (*NEG*). Also known as *TNR* (true negative ratio).

$$Specificity = \frac{TN}{TN + FP}. \tag{26}$$

The area under the ROC curve (*AUC*)—a measure of how well a model can distinguish between two classes and is expressed as follows:

$$AUC = \int_0^1 (TPR)d(FPR) \tag{27}$$

where false positive ratio $FPR = 1 - specificity$.

The Gini index is an alternative to AUC, which is more popular in the context of bankruptcy prediction. *Gini* is a metric that indicates the model's discriminatory power. Moreover, the simple expression of *Gini* is:

$$Gini = 2AUC - 1. \tag{28}$$

The proof of formula transformation from Eqs. (10) to (28) can be found in the articles (*Kaymak, Ben-David & Potharst, 2012*; *Liu et al., 2018*).

Accuracy (*ACC*)—the proportion of correctly classified instances. However, under highly imbalanced problems, it is simple to achieve high accuracy (*Fernández et al., 2018*).

$$ACC = \frac{TP + TN}{TP + TN + FP + FN} \tag{29}$$

$F - score$ is the harmonic mean of precision and recall, where the most common value of $\beta$ is 1. Therefore, the estimate is often called $F1$ or F1 score. However, this measurement focuses on the analysis of the positive class (*Fernández et al., 2018*).

$$F - score = \frac{(1 + \beta^2) Precision \cdot Recall}{\beta^2 \cdot Precision + Recall}. \tag{30}$$

Different from *F1*, the *G–mean* focuses on the balance between instances of majority and minority classes (*Fernández et al., 2018*); it is a geometric mean of the true positive rate and the true negative rate.

$$G - mean = \sqrt{TPR \cdot TNR}. \tag{31}$$

## RESEARCH RESULTS

In this study, experiments to explore the optimal feature set for financial distress classification were performed. Also, 15 different dimensionality reduction techniques and 17 machine learning methods, including their modifications, were analyzed. Each dimensionality reduction technique with a different number of features has been tested ~28 times. In total, 1,433 experiments have been conducted. However, in cases when the classification model assigns all observations to one class, further analysis of the outcomes has not been performed. Thus, the model is required to correctly classify at least half of the negative and positive outcomes. Concerning the imbalance in the test sample, we must correctly classify at least $\geq 1,000$ cases of financial distress and $\geq 35,000$ cases of non-financial distress. This requirement has reduced the number of total experiments by 25% (to 1,071). Figure 4 shows the percentage of cases in each category that meet this assumption. This means, a total of 933 have performed experiments using the feature selection approach, but only 805 (86.26%) have satisfied the stated conditions. All other figures are also based on the aforementioned requirement.

Figure 5 shows the resulting statistics of the top 100 models of the most accurate models according to the *AUC* metric, which balances the recognized financial distress and non-financial distress cases. Results show that the highest *AUC* score is most often achieved

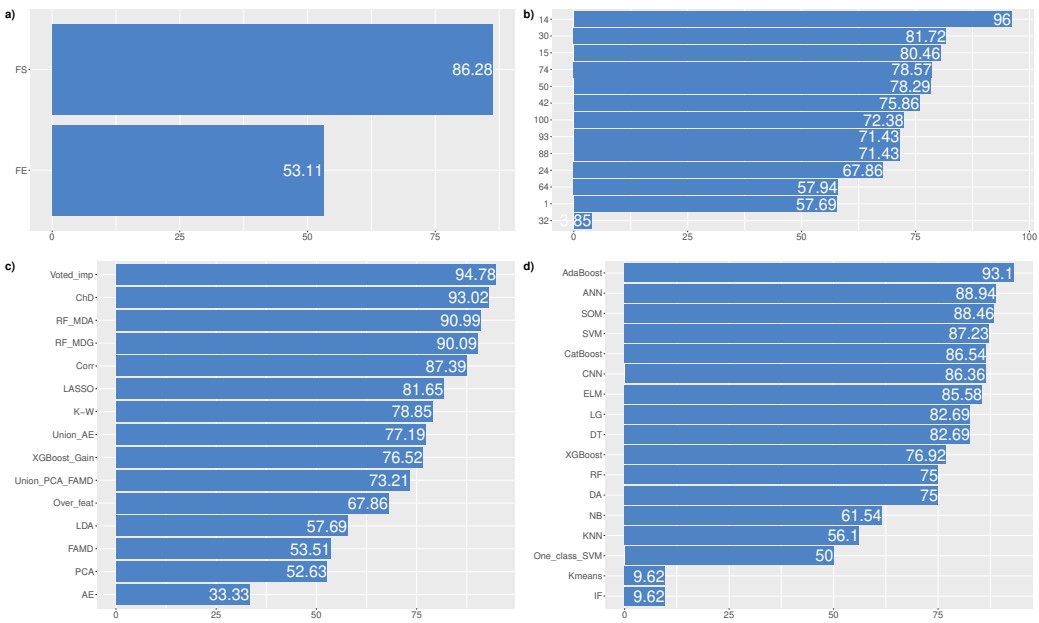

Efficient experiments outcomes for *a*) category of dimensionality reduction; *b*) number of features; *c*) methods of dimensionality reduction; *d*) models of classification.

**Figure 4** **(A–D) Percentage of cases when the model correctly classified at least half of the negative and positive outcomes.**

with the feature selection approach when selecting from 15 to 50 features, using *RF_MDA* methods for dimensionality reduction and ANN models for classification. In contrast, it is observed that the following dimensionality reduction techniques are not worth further investigation: *Overfeat*, *LDA*, *PCA*, *FAMD*, *AE*. Upon analysis of classification techniques, unsupervised methods and KNN, it has been observed that NB algorithm typically yield the lowest *AUC* results, thereby indicating their inefficiency. Therefore, it is not deemed necessary to incorporate them in subsequent analyzes (see Figs. 4 and 5).

Table 4 is formed for the analysis of the first research question –RQ1:which feature selection or extraction method is the most efficient for financial distress identification. This table presents 30 models with the highest performance according to the *AUC* metric. The majority of methods (83.3%), *i.e.,* LASSO, RF or XGBoost, use feature selection embedded techniques. Of these 83.3% the RF method is used as a feature selection algorithm 53.3% times with MDA and MDG features occurring ∼16.7% and ∼36.7%, respectively.

Analyzing Table 4, we can observe that commonly (90% of cases), the highest performance is achieved using neural network models, *i.e.,* ANN (80%) or CNN (10%) methods. Thus, it can be noted that, a simple neuron network structure provides more accurate results. *AUC*, *GINI*, and *G−mean* values show that the highest results are achieved with RF_MDG → 30 ranked features → ANN. However, if we focus on the *AUC* metrics as a balance of recognition between financial distress and non-financial distress cases, as well as on identifying financial distress enterprises a little bit better, we would recommend

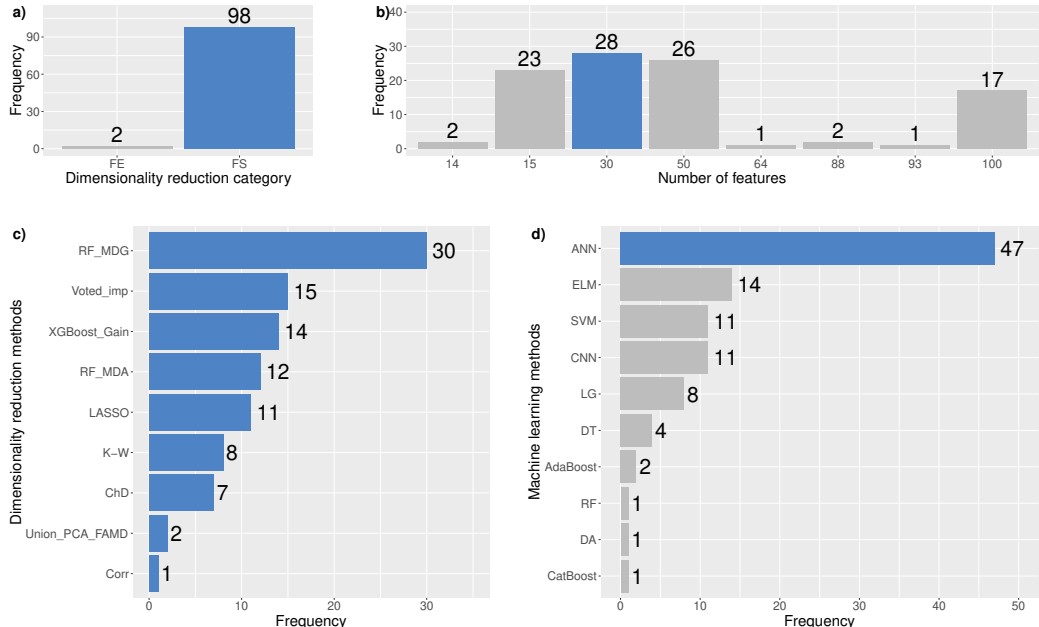

Comparison of *a)* category of dimensionality reduction; *b)* the used number of features; *c)* methods of dimensionality reduction; *d)* models of classification.

**Figure 5** (A-D) Top 100: The best combinations of methods according to *AUC*.

choosing XGBoost_Gain → 50 ranked features → ANN_I. This results in the specificity metric increasing by 5.42% while *AUC* decreases by only 1.68%.

On the contrary, if we focus on the non-financial distress identification, using CNN_I instead of ANN_I is recommended, since it has a higher recall metric by 3.83% and *AUC* becomes lower only by 1.8%. The order of priority for the classes is crucial: first, a solid recognition of both classes is necessary, followed by the classification of enterprises in financial distress or not. Otherwise, models that assign the majority (or all) of the outcomes to a single class will be given precedence.

## A benchmark model

A benchmark financial distress model has been added to the research to improve comparability with the proposed methodology. The logistic regression approach employing financial ratios data serves as a benchmark model, since it is the most commonly employed approach in research on financial distress or bankruptcies (*Kuiziniene et al., 2022*).

Foremost, the benchmark model has been implemented in correlation analysis. If two features are highly correlated ($> 0.7$), then the feature with a lower correlation to the target is removed. In the logistic regression, the Wald test is commonly used to identify statistically significant variables. The selected significance level for this research is 0.05. Features used in the benchmark model are in bold and underlined (see Appendix A, Table A.3).

The benchmark model's results are presented in Tables 5 and 4. The confusion matrix of the benchmark model shows that the effectiveness criteria are fulfilled – the model

**Table 4  Performance of classification models using different dimensionality reduction techniques.**

| # | DRC[*] | DRM[**] | No.[***] | Method | Accuracy | AUC | F–1 | G–mean | Gini | Precision | Recall | Specificity |
|---|---|---|---|---|---|---|---|---|---|---|---|---|
| 1. | **FS** | **RF_MDG** | **30** | **ANN** | 0.9038 | **0.8687** | 0.9482 | **0.8679** | **0.7375** | 0.9946 | 0.9059 | 0.8316 |
| 2. | FS | RF_MDG | 30 | ANN_I | 0.9028 | 0.8682 | 0.9476 | 0.8675 | 0.7365 | 0.9946 | 0.9048 | 0.8316 |
| 3. | FS | RF_MDG | 50 | ANN | 0.8975 | 0.8679 | 0.9446 | 0.8673 | 0.7358 | 0.9948 | 0.8992 | 0.8365 |
| 4. | FS | RF_MDG | 30 | ANN_II | 0.9184 | 0.8657 | 0.9564 | 0.8639 | 0.7314 | 0.9941 | 0.9215 | 0.8099 |
| 5. | FS | RF_MDG | 50 | ANN_I | 0.8988 | 0.8607 | 0.9454 | 0.8597 | 0.7213 | 0.9943 | 0.9010 | 0.8203 |
| 6. | FS | Voted_imp | 50 | ANN | 0.8924 | 0.8595 | 0.9417 | 0.8588 | 0.7191 | 0.9944 | 0.8944 | 0.8247 |
| 7. | FS | RF_MDG | 30 | ANN_III | 0.8881 | 0.8588 | 0.9392 | 0.8582 | 0.7175 | 0.9944 | 0.8899 | 0.8277 |
| 8. | FS | LASSO | 50 | ANN | 0.9035 | 0.8571 | 0.9480 | 0.8557 | 0.7142 | 0.9939 | 0.9062 | 0.8080 |
| 9. | FS | LASSO | 30 | ANN_I | 0.9264 | 0.8565 | 0.9609 | 0.8533 | 0.7129 | 0.9933 | 0.9305 | 0.7824 |
| 10. | FS | RF_MDA | 100 | ANN_I | 0.9052 | 0.8561 | 0.9490 | 0.8545 | 0.7122 | 0.9938 | 0.9081 | 0.8040 |
| 11. | FS | LASSO | 30 | ANN_III | 0.9061 | 0.8558 | 0.9495 | 0.8541 | 0.7116 | 0.9937 | 0.9091 | 0.8026 |
| 12. | FS | RF_MDA | 30 | ANN | 0.9138 | 0.8548 | 0.9539 | 0.8525 | 0.7096 | 0.9935 | 0.9173 | 0.7922 |
| 13. | FS | XGBoost_Gain | 50 | CNN_II | 0.8457 | 0.8537 | 0.9141 | 0.8536 | 0.7073 | 0.9953 | 0.8452 | 0.8621 |
| 14. | FS | XGBoost_Gain | 15 | ANN | 0.8919 | 0.8533 | 0.9414 | 0.8523 | 0.7066 | 0.9940 | 0.8942 | 0.8124 |
| 15. | FS | RF_MDG | 50 | SVM radial | 0.9184 | 0.8526 | 0.9565 | 0.8498 | 0.7052 | 0.9932 | 0.9224 | 0.7829 |
| 16. | **FS** | **XGBoost_Gain** | **50** | **ANN_I** | 0.8200 | 0.8519 | 0.8983 | 0.8512 | 0.7038 | **0.9960** | 0.8181 | **0.8858** |
| 17. | FS | Voted_imp | 100 | ANN_I | 0.9008 | 0.8514 | 0.9466 | 0.8498 | 0.7029 | 0.9936 | 0.9038 | 0.7991 |
| 18. | FS | RF_MDA | 50 | ANN | 0.9277 | 0.8512 | 0.9617 | 0.8473 | 0.7024 | 0.9929 | 0.9323 | 0.7701 |
| 19. | FS | RF_MDA | 30 | ANN_I | 0.9058 | 0.8509 | 0.9494 | 0.8489 | 0.7018 | 0.9934 | 0.9091 | 0.7927 |
| 20. | **FS** | **XGBoost_Gain** | **50** | **CNN_III** | **0.9389** | 0.8507 | **0.9678** | 0.8456 | 0.7014 | 0.9926 | **0.9442** | 0.7573 |
| 21. | FS | Voted_imp | 50 | ANN_I | 0.8692 | 0.8502 | 0.9282 | 0.8500 | 0.7005 | 0.9944 | 0.8704 | 0.8301 |
| 22. | FS | RF_MDA | 30 | ANN_II | 0.9064 | 0.8498 | 0.9497 | 0.8477 | 0.6996 | 0.9933 | 0.9098 | 0.7898 |
| 23. | FS | RF_MDG | 30 | SVM radial | 0.9131 | 0.8494 | 0.9535 | 0.8467 | 0.6988 | 0.9932 | 0.9169 | 0.7819 |
| 24. | FS | LASSO | 50 | ANN_I | 0.9190 | 0.8493 | 0.9568 | 0.8461 | 0.6987 | 0.9930 | 0.9232 | 0.7755 |
| 25. | FS | LASSO | 30 | ANN_II | 0.9278 | 0.8493 | 0.9617 | 0.8452 | 0.6986 | 0.9928 | 0.9324 | 0.7661 |
| 26. | FS | K–W | 100 →20 | LG | 0.8375 | 0.8492 | 0.9092 | 0.8491 | 0.6985 | 0.9952 | 0.8368 | 0.8616 |
| 27. | FS | RF_MDG | 100 | ANN | 0.8645 | 0.8488 | 0.9255 | 0.8486 | 0.6976 | 0.9944 | 0.8655 | 0.8321 |
| 28. | FS | RF_MDG | 100 | ANN_I | 0.8704 | 0.8487 | 0.9289 | 0.8484 | 0.6974 | 0.9942 | 0.8717 | 0.8257 |
| 29. | FS | K–W | 100 | CNN_III | 0.8143 | 0.8483 | 0.8947 | 0.8475 | 0.6965 | 0.9959 | 0.8122 | 0.8843 |
| 30. | FS | RF_MDG | 50 | ANN_II | 0.8758 | 0.8481 | 0.9321 | 0.8476 | 0.6963 | 0.9940 | 0.8775 | 0.8188 |
| 984. | | Benchmark | 56 →51 | LG | 0.8070 | 0.6991 | 0.8912 | 0.6898 | 0.3984 | 0.9854 | 0.8135 | 0.5832 |

**Notes.**
 [*]Dimensionality reduction category.
 [**]Dimensionality reduction methods.
 [***]The number of used features.
 The best models and their performance values are shown in bold.

**Table 5  A confusion matrix of the benchmark model.**

| | | Predicted class | | |
|---|---|---|---|---|
| | | Non-financial distress | Financial distress | Total |
| Actual class | Non-financial distress | 56,994 | 13,070 | 70,064 |
| | Financial distress | 843 | 1,180 | 2,023 |
| | Total | 57,837 | 14,250 | 72,087 |

correctly classified 1,180 Financial distress cases and 56,994 Non-financial distress cases (see Table 5). However, the *AUC* score was found to be only 0.6991. The benchmark model is outperformed by the combinations of methods and application techniques proposed in the methodology. Moreover, when ranked by the *AUC* metric, the benchmark model is ranked 984 out of 1071 experiments (see Table 4).

## Feature selection

In this section, only the results of the feature selection approach and the analysis of two research questions are provided, *i.e.,* RQ2–which data categories are the most important in financial distress prediction? and RQ3–does the composition of the final feature set depend on the feature selection technique?

Figure 6A shows the *AUC* results for the best machine learning models for each feature selection technique with a certain number of features, *i.e.,* constant parameters are: feature selection technique and the number of features. To give an example, 26 experiments were done using the *RF_MDG* feature selection technique with 30 significant features. The one with the highest *AUC* score was found to be 0.8687. The best *AUC* results for ML models with different numbers of features are distributed between 0.8012 and 0.8687 for all used FS methods with different features. Embedded and hybrid feature selection methods achieve a relatively high accuracy, *i.e., AUC* score ≥ 0.85. Therefore, future experiments should be conducted with these groups on the topic of financial distress. Wrapper methods in this research have not been analyzed due to the long estimation time and limited computation resources. In further research, this method could be implemented after the embedded method feature ranking and the selection of an initial number of features. This study indicates that the best results are achieved when using sets of up to 50 features or fewer. Also, using simple (not deep) neural network methods to recognize financial distress in enterprises is found to give the best results.

Figure 6B shows how much each data category (see Table 2) is used in respect to the other categories. It can be observed that all FS methods use features from *Financial statements*, *Social Insurance* and *Other* category. Contrary to this, *Macro*, *Enterprise type* and *Sector's* data categories are rarely selected and only by a few FS methods. A comparison of filter and embedded FS techniques shows that the filter methods avoid using features from the *Taxes* data category.

Different FS methods tend to select features from the same data category as the number of features increases, *e.g., Social insurance* could be split into employees and delay of taxes (including delay in days and amount information). In that case, it can be seen that if FS

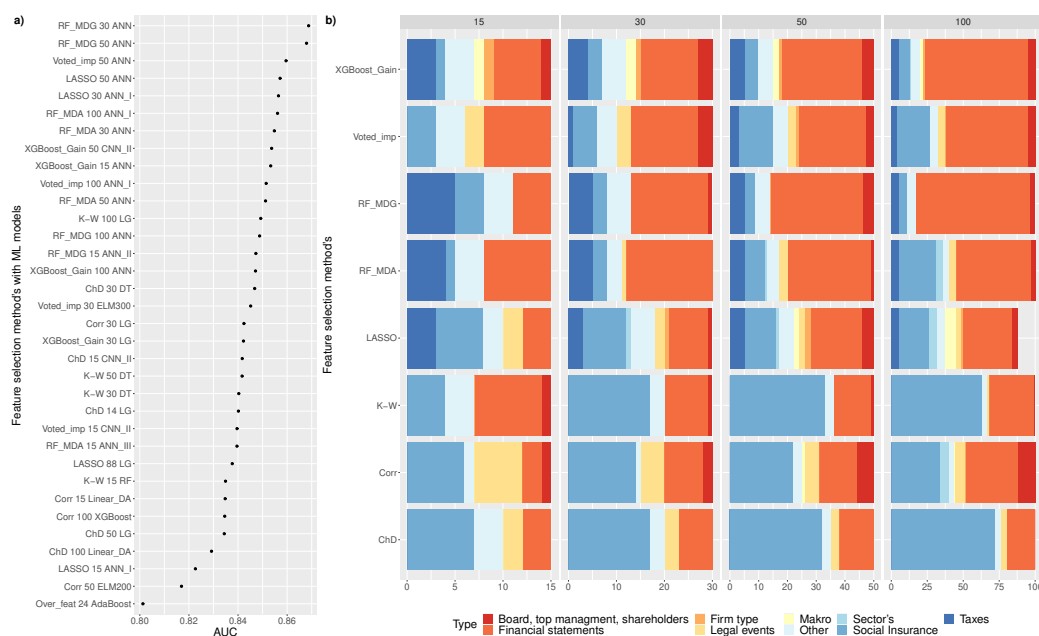

*a*) Ranked *AUC* results by the best ML model for all used FS methods with different numbers of features. All presented ML methods outperformed the benchmark score (*AUC*=0.6991); *b*) Comparison between the FS method and the used feature data category.

**Figure 6**  **Comparison of feature selection methods.**

methods choose *Social insurance* → employees subcategory, increasing the number of the features would include more features from this particular subcategory.

## Feature extraction

This section presents the results from feature extraction experiments as well as the analysis of the research question—RQ4: which feature extraction strategy is more efficient: analyzing all features at once *vs* analyzing features by categories?

The idea represented in Fig. 7A is the same as for Fig. 6A, but it is intended for the analysis of feature extraction experiments. Upon comparing the results obtained with FE and FS (see Figs. 6A and 7A), it is observed that FE methods' superior *AUC* results are distributed lower than those obtained from FS, ranging from 0.6513 to 0.8381 (see Fig. 7A). In addition, the best results for the AE feature extraction technique, with 32 and 100 relevant features, were found to be lower than the benchmark model. The highest *AUC* ≥ 0.83 from the FE methods category has been achieved with the implementation of the union strategy for PCA_FAMD and AE methods. This also resulted in the highest number of used features in PCA and FAMD methods.

In this study, the union of separate autoencoders (see the 'Dimensionality reduction methods' section; *Union_PCA_FAMD* and *Union_AE*) has been implemented for both linear and non–linear data transformations separately, with respect to the analyzed data category. This approach involves the creation of data categories with reduced dimensionality and their connection to a single feature space. The usage of the Union strategy for linear and non–linear data transformation has given the best *AUC* score when compared to

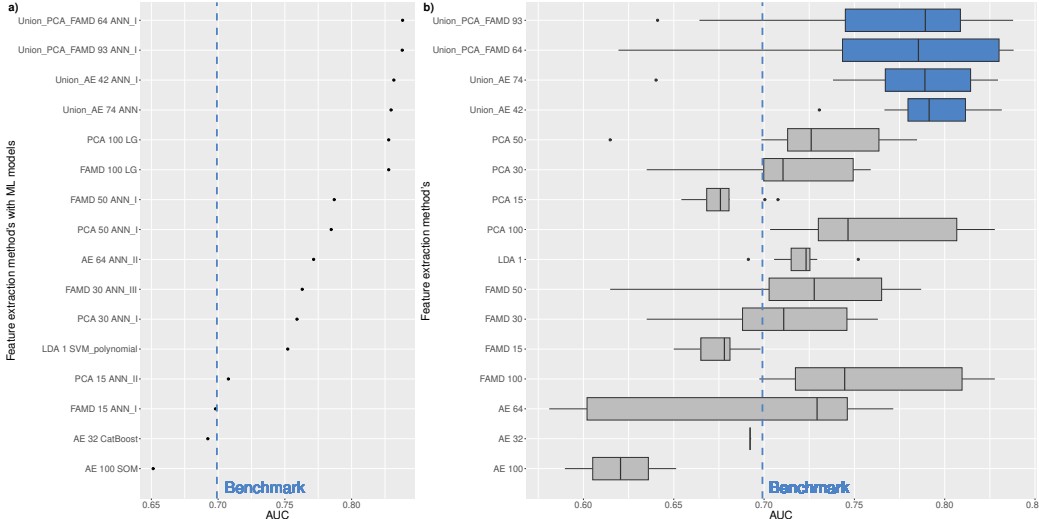

*a*) Ranked *AUC* result by best ML model for all used FE methods with different numbers of features; *b*) Comparison of all ML models' *AUC* results for all used FE methods with different numbers of features.

**Figure 7** Comparison of feature extraction methods performance.

all ML methods separately. According to the distributions presented in Fig. 4A, only 53.11% of the experiments were completed using FE methods because the minimum classification quotas were not reached. Figure 7B shows the *AUC* score distribution of ML models using FE methods. Moreover, *Union_PCA_FAMD* and *Union_AE* achieved efficient results—73.21% and 77.19%, respectively (see Fig. 4C). This confirms the union of separate autoencoders strategy's viability and proves that its development in further research could lead to better results. In addition, better Union results are achieved when a smaller number of features is present, rather than a larger one. Moreover, a linear transformation provides better results than a non–linear one.

## Machine learning models

This part presents the analysis of the last research question—RQ5: which machine learning model performs better in financial distress identification?

In this study, experiments with 17 different machine learning models and their 28 modifications have been provided. However, not all experiments have succeeded due to the time limit for a single method (24 h). Usually, the cycle of all ML models, for the one-dimensionality reduction method, takes from 2 to 4 days. A comparison of models and their modifications is presented in Fig. 8. Blue color indicates the better models. Comparing dimensionality reduction categories, *i.e.,* FE and FS, the efficiency trends remain the same between models. However, higher *AUC* score is achieved in the FS approach. What concerns the comparison of tree-based algorithms (Fig. 8C), it is difficult to select the most effective methods as their *AUC* scores are similar. Moreover, in the FS

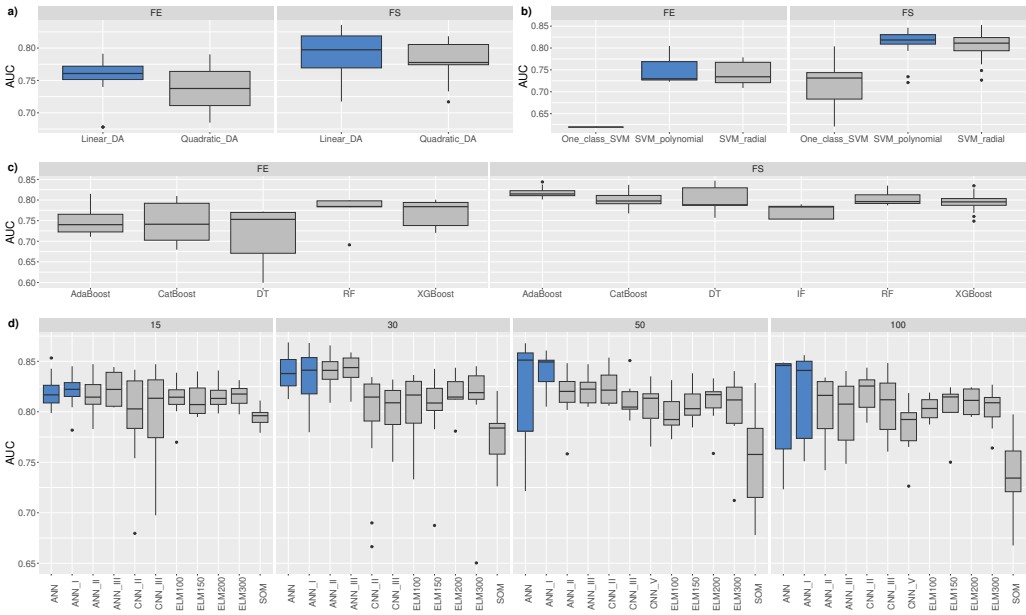

a) performance of discriminant analysis models; b) performance of SVM models; c) performance of tree-based algorithms; d) performance of neural network methods in FS approach.

**Figure 8** (A–D) Comparison of classification methods performance by *AUC* score.

approach the achieved *AUC* score variance is small, despite the used FS method, which shows the stability of the ML method and equally important feature selection.

As the FS approach has a similar tendency, a larger sample size with more significant variables, and their number of included features in the analysis, the comparison of neural network methods is given. The main conclusion remains the same – simpler (not deep) neural network methods give better results in recognition of enterprises that have financial distress. However, the more features are included in the analysis, the less stable the neural network results become if all conditions remain the same, except when the FS method is used. This implies that the FS method is important. In further research, it is necessary to expand the neural network structure searches for each dimensionality reduction method separately in order to achieve a deeper comparison of the results.

## CONCLUSIONS

This study presents a methodology for feature selection and feature extraction methods in financial distress identification. The main goal is to identify a subset of features which lead to the minimization of the loss function as an estimate of belonging to financial distress in the enterprise. For the benchmark model, the logistic regression method with financial ratios data was used. Its *AUC* score resulted in 0.6991 and the $G-mean$ was equal to 0.6898. Also, 983 of the models suggested in the methodology have outperformed the benchmark model. In this comparative study, the highest results (*AUC* (0.8687), *GINI* (0.7375), and $G-mean$ (0.8679)) were achieved using a feature selection embedded technique – random forest with mean decreasing Gini → 30 ranked features → artificial neural network.

However, if a slightly better majority class recognition is more important than the balance of classes, then the recommended techniques are different. In this case, changing to the feature selection embedded technique – Extreme Gradient Boosting Machine with Gain metrics → 50 ranked features → artificial neural network with one hidden layer, increased *specificity metric* by 0.0542 (0.8858), and decreased *AUC* by only 0.0168 (Gini 0.0337). The best classification results are provided by the neural network method, which the authors (*Tsai et al., 2021*) suggest can be improved by adding the Bagging algorithm. Additionally, the research combines the findings of diverse authors who have identified features within a single dataset (*Hafeez & Kar, 2018*; *Fernández-Gámez et al., 2020*; *Jindal, 2020*; *Rezende et al., 2017*; *Jones, 2017*; *Ashraf, Félix & Serrasqueiro, 2021*; *Faris et al., 2020*; *Liang et al., 2020*; *Volkov, Benoit & Van den Poel, 2017*), resulting in the formation of 972 dimensions. Comparing the width of the analyzed studies, it is noticeable that the researchers used FS or FE approach to reduce the sample from 27–170 to 7–21 features (*Séverin & Veganzones, 2021*; *Perboli & Arabnezhad, 2021*; *Ben Jabeur, Stef & Carmona, 2022*; *Altman et al., 2022*; *Zizi et al., 2021*; *Du et al., 2020*; *Acharjya & Rathi, 2021*; *Soui et al., 2020*). In our study, a total of 972 features have been reduced to 15–100 features, resulting in a reduction of 1.54% to 10.29% of the entire data set. What concerns data reduction, different feature selection and extraction techniques were used, which were chosen based on data types, running time, overfitting, and suitability for unseen data.

Our additional findings are as follows:

1. Feature selection approach results proved the suitability of embedded and hybrid methods for financial distress identification. The best results were obtained when analyzing feature sets consisting of ≤50 features. Commonly, features in FS methods were selected from Financial statements and Social Insurance data categories.

2. Feature extraction approach results approved the usage of the union strategy. The union strategy has been implemented separately for linear and non–linear data transformations based on the data category. This led to dimensionality-reduced data categories that are linked to a single feature space, *e.g.*, constructing 12 unique autoencoders for each data category and then combining them into one feature set. The highest *AUC* score was achieved after the implementation of the union strategy in the FE approach for all ML separately.

3. The analysis of the implementation of machine learning methods for identifying financial distress reveals that simpler (not deep) neural network methods give better results. Furthermore, in order to conduct a more comprehensive comparison of the findings, it is imperative to extend the neural network structure experiments for each dimensionality reduction technique individually in further research. Moreover, the findings indicate that the unsupervised methods that have been analyzed are not worthy of further investigation.

There are a few noteworthy limitations to this study.

1. Test set. The research results were tested based on the information available for half a year.

2. Number of features. Feature ranking has been used after the implementation of feature selection techniques. However, it is still unclear how to select the appropriate number

of features. Therefore, the number of features has been randomly selected to make the methods comparable.

3. Stability of feature set. The stability of the chosen features over time was not examined.
4. Balancing technique. It is unclear how different data balancing techniques would affect the results. In this study, only one balancing technique was used.
5. Model specifications. Model parameter optimization was not performed; baseline models were used to compare results. However, small changes in the results of the neural network models were observed when changing the number of features.

In further research, we plan to focus on the implementation of dimensionality reduction techniques from a ranked feature stability point of view when time changes. Moreover, we consider performing experiments using the combination of feature selection and extraction techniques for different data categories.

## ACKNOWLEDGEMENTS

We wish to thank Viktoras Vaitkeviçius from Baltfakta for providing data and fruitful discussions and Arnas Matusevçius from VMU and CARD for technical support.

### Funding

This study has received funding under the Horizon Europe Widening Participation program - Teaming for Excellence 2022 (Centre of Excellence of AI for Sustainable Living and Working (SustAInLivWork) project, grant agreement No. 101059903) and from the European Union. The funders had no role in study design, data collection and analysis, decision to publish, or preparation of the manuscript.

### Grant Disclosures

The following grant information was disclosed by the authors:
Horizon Europe Widening Participation program - Teaming for Excellence 2022 (Centre of Excellence of AI for Sustainable Living and Working (SustAInLivWork) project) and from the European Union: 101059903.

### Competing Interests

Robertas Damaševičius is an Academic Editor for PeerJ.

### Author Contributions

- Dovilė Kuizinienė conceived and designed the experiments, performed the experiments, analyzed the data, performed the computation work, prepared figures and/or tables, authored or reviewed drafts of the article, and approved the final draft.
- Paulius Savickas conceived and designed the experiments, performed the experiments, analyzed the data, performed the computation work, prepared figures and/or tables, and approved the final draft.

- Rimantė Kunickaitė conceived and designed the experiments, performed the experiments, analyzed the data, performed the computation work, prepared figures and/or tables, and approved the final draft.
- Rūta Juozaitienė conceived and designed the experiments, performed the experiments, analyzed the data, performed the computation work, prepared figures and/or tables, and approved the final draft.
- Robertas Damaševičius analyzed the data, prepared figures and/or tables, authored or reviewed drafts of the article, and approved the final draft.
- Rytis Maskeliūnas analyzed the data, authored or reviewed drafts of the article, and approved the final draft.
- Tomas Krilavičius conceived and designed the experiments, authored or reviewed drafts of the article, and approved the final draft.

## Data Availability

The dataset is available at Figshare: Damasevicius, Robertas (2023). Experimental data for paper "A comparative study of feature selection and feature extraction methods for financial distress identification". figshare. Dataset. https://doi.org/10.6084/m9.figshare.23507826.v1.

## Supplemental Information

Supplemental information for this article can be found online at http://dx.doi.org/10.7717/peerj-cs.1956#supplemental-information.

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
