# Peer review of "A comparative study of feature selection and feature extraction methods for financial distress identification"

_PeerJ Computer Science, doi:10.7717/peerj-cs.1956_

## Round 0.1 · original submission · Major Revisions

Dear authors,

Thank you for your submission. Your article has not been recommended for publication in its current form. However, we do encourage you to address the concerns and criticisms of the reviewers and resubmit your article once you have updated it accordingly. Especially following issues and concerns should be addressed:

1- The study would be more useful if it included a comprehensive discussion of the criteria used to select the particular methods of feature selection and extraction that were analysed.

2- The paper makes no explicit mention of overfitting, which occurs when a model is overly sophisticated and fits the noise in the data rather than the underlying patterns.

3- The study lacks a clear explanation for why the particular financial distress identification model was chosen as the benchmark for comparison.

Best wishes,

**Language Note:** PeerJ staff have identified that the English language needs to be improved. When you prepare your next revision, please either (i) have a colleague who is proficient in English and familiar with the subject matter review your manuscript, or (ii) contact a professional editing service to review your manuscript. PeerJ can provide language editing services - you can contact us at [email protected] for pricing (be sure to provide your manuscript number and title). – PeerJ Staff

·

Basic reporting

The English used throughout was clear enough. The concepts described are complex, as are the figures, but enough description is given to decipher what they did.
The literature references seem valid enough - there are many of them.
The article structure was fine.
The results, as stated above, are complex, but with effort the reader can figure out results.

Experimental design

A complex design due to attacking so many dimensions. The paper gives a pretty comprehensive framework of variable selection, balancing, and algorithm selection. The methodology was thoroughly described.
One issue I have is the definition of decision trees in section 4.3 1 c - you describe decision trees in the context of decision analysis. In the context of data mining, they are a data mining classification method in which the outcomes are described in a tree. Random forests, which turned out to be quite good in your experiments, use a multitude of decision tree runs. (XBoost also tends to do well, as you also found.).

Validity of the findings

I think it is a good article meriting publication - given that you fix section 4.3 1 c.
It probably is more complex in research design than necessary.

Additional comments

The paper is a thorough effort.

·

Basic reporting

Overall, the paper examines the use of feature selection tools and machine learning models for predicting bankruptcy in Lithuania. While the experiment is extensive, the contribution in both Computer Science and Finance is only partial and should be explicitly stated. Additionally, there are several areas that need improvement:
1. Table 1 would be viewed in a transpose form.
2. The evidence for Filter techniques is weak, with only one example provided for each case. The defense for Relief-f raises doubts about its consideration.

Experimental design

3. When describing data, the justification for the restrictions on firms is missing. Please provide a reason for this.
4. The second class condition should not use "etc" as it is a definition and all cases should be clearly stated.
5. The financial statement variables/indicators used should be presented.
6. The explanation of Figure 1 in paragraph in lines 324-334 is inadequate. Improve this section by clarifying vote-based FS, how votes were evaluated, and the meaning of "Y" in the figure. I suggest to define each variable separately.
7. Clarify how normalization was performed, whether by year or on the full dataset, as it is unclear in both the text and algorithm.
8. The 5th step needs clarification on why Unsupervised ML does not require balanced data and the unclear obligation of SMOTE-NC.

Validity of the findings

9. Although have answered all proposed questions in the results section, please, engage with existing literature to make comparisons and highlight advancements and improvements.
10. Consider presenting descriptive statistics about all variables in a supplemental file.

Additional comments

Minor review points:
1. Review citation style for Verma (2020) in both citations (wrong style) and reference list (missing data).
2. Check for parentheses used incorrectly, such as in Table 1. Please, review all.
3. Ensure there is a space between text and parentheses throughout the paper. E.g., "regression trees(Azayite and Achchab, 2018)" in line 154. Please, review all.
4. Review and correct any typos, such as inverted quotation marks (for instance, ”good”) and equation 6.

Reviewer 3 ·

Basic reporting

The authors use a rigorous and systematic approach to evaluate the performance of the different feature selection and feature extraction methods, which enhances the reliability and validity of the results.
The study includes a detailed description of the data preprocessing steps, such as normalization and outlier detection, which helps to ensure the quality and consistency of the data.

Experimental design

The authors use a variety of performance metrics, such as accuracy, precision, recall, and F1 score, to evaluate the effectiveness of the different methods, which provides a more comprehensive and nuanced assessment of their performance.

Validity of the findings

The study could benefit from a more detailed discussion of the criteria used to select the specific feature selection and feature extraction methods that were evaluated. For example, the authors do not explain why they chose to include certain methods and exclude others.
The study does not explicitly address the issue of overfitting, which can occur when a model is too complex and fits the noise in the data rather than the underlying patterns. The authors could have used techniques such as cross-validation or regularization to mitigate this issue.
The study does not provide a clear rationale for the choice of the specific financial distress identification model that was used as the benchmark for comparison.
The authors could have compared the performance of the different methods against multiple benchmarks to provide a more robust evaluation.

---

## Round 0.2 · Major Revisions

Dear authors,

Thank you for revised article. A reviewer who accepted the invitation to review your article has not yet submitted their suggestions. Reviewer 1 has now commented on your article and suggests major revisions. We encourage you to address the reviewer's concerns and criticisms and resubmit your article once you have updated it accordingly.

Best wishes,

·

Basic reporting

While the grammar and writing style are generally acceptable, there are a few areas that could be improved. For example, the sentence ‘The advancements in information technology and the escalating volume of stored data have led to the emergence of financial distress that transcends the realm of financial statements and derivative indicators (ratios)’ could be misinterpreted. It might seem as though IT and big data are causing financial distress. Also, the term ‘derivative’ can be confusing in a financial context due to its association with a specific market.

Some sentences are quite lengthy, which can make them hard to comprehend. It would be beneficial to break them down into shorter, more straightforward sentences.

The first paragraph in the Literature Review Section is somewhat unclear. It begins with a discussion on financial distress, then abruptly shifts to machine learning techniques, and concludes with an explanation of dimensionality reduction. This section could benefit from a review.

It’s recommended to avoid bolding body text (as seen on p. 5, lines 118 and 124), and to refrain from using symbols (for example, ~3% on p.6, l. 172).

Experimental design

The response to Reviewer 1’s first query seems insufficient. You’ve mentioned enhancing the descriptions of the figures, but the captions for all figures still lack clarity. For instance, Figure 1 needs more detailed information about the measures used (scale, size, etc). In Figure 2, it’s unclear what 4a, 4b, and 4c represent. Figures 4-8 each have two captions, which is confusing.

The evidence supporting the use of Filter techniques is not robust enough.

On page 8, line 286, you’ve stated that the filter was applied for “stability”. Could you please elaborate on what this stability entails?

The features are now satisfactory. However, considering the numerous ratios and definitions, it would be beneficial to provide them in a repository. The normalization process is acceptable.

The fifth step appears to be a bit confusing due to the mixture of requirements from both supervised and unsupervised methods.

Validity of the findings

The discussion of findings in relation to the existing literature appears to be insufficient yet. Although you intended to engage more thoroughly with the literature in the results section, it seems that the specific comparisons, advancements, and improvements were not adequately highlighted. Could you please point out the specific pages and lines where these elements are discussed?

Additional comments

The minor errors now appear to be nonexistent.

---

## Round 0.3 · Major Revisions

Dear authors,

Thank you for submitting your revised article. Reviewers have now commented on your article. Your article has not been recommended for publication in its current form. We encourage you to clearly address the concerns and criticisms raised by Reviewer 2 and resubmit your revised article once you have updated it accordingly.

Best wishes,

·

Basic reporting

The revision is clear enough.

Experimental design

It is valid enough - fully explained.

Validity of the findings

Such studies always find their results best - but there are many other methods that work as well for specific datasets.

Additional comments

Go ahead.

·

Basic reporting

1) "Some sentences are quite lengthy, which can make them hard to comprehend. It would be beneficial to break them down into shorter, more straightforward sentences.
AUTHORS: Thank you, we have revised accordingly."
>> In the Introduction, this issue persists.

2) "The first paragraph in the Literature Review Section is somewhat unclear. It begins with a discussion on financial distress, then abruptly shifts to machine learning techniques, and concludes with an explanation of dimensionality reduction. This section could benefit from a review.
AUTHORS: We have expanded as requested.""
>> First paragraph ok.

3) "It’s recommended to avoid bolding body text (as seen on p. 5, lines 118 and 124), and to refrain from using symbols (for example, ~3% on p.6, l. 172).
AUTHORS: We have revised."
>> OK

Experimental design

1) "The response to Reviewer 1’s first query seems insufficient. You’ve mentioned enhancing the descriptions of the figures, but the captions for all figures still lack clarity. For instance, Figure 1 needs more detailed information about the measures used (scale, size, etc). In Figure 2, it’s unclear what 4a, 4b, and 4c represent. Figures 4-8 each have two captions, which is confusing.
AUTHORS: We have revised:
Fi"gure 1 – change to a new plot
Figure 2 – Explained in the fourth step, with links to 4a, 4b, and 4c
Figures 4-8 – each figure has only one caption, however, they consist of several graphs, a part of each graph is explained separately (as a, b, c part)."

>> Figures OK

2) "The evidence supporting the use of Filter techniques is not robust enough.
AUTHORS: All literature analyses articles are based on the applicability of these techniques in solving the FD problem."
>> The authors do not present any argument and the text remains the same. You should edit the text to give more validity and evidences that these techniques are useful and contributive to solvo the problem.

3) "On page 8, line 286, you’ve stated that the filter was applied for “stability”. Could you please elaborate on what this stability entails?
AUTHORS: The line is removed, everything is said before: enterprise has provided at least one financial statement from the last two years""
>> I did not understand this answer. There are no information about stability in this version. The problem is: what is the definition of stability in this context?

4) "The features are now satisfactory. However, considering the numerous ratios and definitions, it would be beneficial to provide them in a repository. The normalization process is acceptable.
AUTHORS: Before that, there was a comment that we need to add in appendix , if necessary we will upload to the repository (but only the data description).
It would be better not to do this, because as I have checked many repositories of such articles, most of the links do not work."
>> You provide a link for data but it's not working. The transparency of this research is compromised.

5) "The fifth step appears to be a bit confusing due to the mixture of requirements from both supervised and unsupervised methods.
AUTHORS: we have added links in the text to 5a and 5b""
>> OK.

Validity of the findings

1) "The discussion of findings in relation to the existing literature appears to be insufficient yet. Although you intended to engage more thoroughly with the literature in the results section, it seems that the specific comparisons, advancements, and improvements were not adequately highlighted. Could you please point out the specific pages and lines where these elements are discussed?
AUTHORS: see lines 784-792."
>> OK. However, this is unusual as discussion should be done before conclusions.

Additional comments

"The minor errors now appear to be nonexistent.
AUTHORS: we have revised accordingly."

OK

---

## Round 0.4 · accepted · Accept

Dear authors,

Thank you for the revision and for clearly addressing all the reviewers' comments. I confirm that the paper is improved and addresses the concerns of the reviewers. Your paper is now acceptable for publication in light of this revision.

Best wishes,

·

Basic reporting

The paper is ok.

Experimental design

ok

Validity of the findings

ok